# DUET: Dual-Perspective Pseudo Labeling and Uncertainty-aware Exploration & Exploitation Training for Source-Free Domain Adaptation

**Jae Yun Lee**[1]    **JaeHyeon Park**[2],    **Gyoomin Lee**[2],    **Bogyeong Kim**[3],
**Min Hee Cha**[3],    **Hyeok Nam**[1],    **Joo Hyeon Jeon**[1],    **Hyunse Lee**[2],
**Sung In Cho**[1]*

[1] Department of Artificial Intelligence at Sogang University
35 Baekbeom-ro, Mapo-gu, Seoul 04107, Republic of Korea
`{stcaft, skagur10, jhbox7, csi2267}@sogang.ac.kr`

[2,3] Department of Computer Science and Artificial Intelligence at Dongguk University
30 Pildong-ro 1-gil, Jung-gu, Seoul 04620, Republic of Korea
[2]`{pjh0011, exceleaf, sae4394}@dongguk.edu`
[3]`{qhruddl51, rninicha}@dgu.ac.kr`

## Abstract

Source-free domain adaptation (SFDA) aims to adapt a pre-trained source model to an unlabeled target domain without requiring labeled source data. In a self-supervised setting, relying on pseudo labels on target domain samples facilitates the domain adaptation performance providing strong supervision. However, a critical problem of this approach is the inherent instability of the pre-trained source model in the target domain, leading to unreliable pseudo labels for the target domain data. To tackle this, we propose a novel Dual-perspective pseudo labeling strategy that jointly leverages a task-specific perspective and a domain-invariant perspective, assigning pseudo labels only to target samples on which the target model's predictions and CLIP's predictions agree. To further enhance representation learning without introducing noisy supervision, we apply consistency training to uncertain samples. Additionally, we introduce a Tsallis mutual information (TMI)-based vision optimization strategy guided by an Uncertainty-based adaptation index (UAI), which dynamically modulates entropy sensitivity based on the model's adaptation uncertainty. The UAI-based training paradigm enables stable and adaptive domain alignment by effectively balancing exploration and exploitation processes during the optimization process. Our proposed method achieves state-of-the-art performance on domain adaptation benchmark datasets, improving adaptation accuracy by 1.6% on Office-Home, 1.4% on VisDA-C, and 2.9% on DomainNet-126, demonstrating its effectiveness in SFDA. The code is publicly available at https://github.com/l3umblee/duet-sfda.

## 1 Introduction

Unsupervised domain adaptation (UDA)[26, 3, 15, 6] explores training strategies or model architectures that enable a model trained on labeled source domain data to achieve high task accuracy on the unlabeled target domain data. Traditionally, UDA assumes access to both labeled source domain data

---

*Corresponding author.

and unlabeled target domain data. However, in practical scenarios, the application of UDA can be limited due to privacy concerns, legal regulations, and data security issues associated with labeled source domain data.

To address these limitations, source-free domain adaptation (SFDA)[12, 1, 28, 29, 30, 38, 11, 13, 27] has been proposed, allowing domain adaptation without the use of labeled source domain data. SFDA relies solely on a pre-trained source model and unlabeled target domain data to achieve adaptation. SFDA can be broadly categorized into two main approaches. The data-based approach[32, 10, 39, 8] focuses on constructing a virtual source domain using adversarial learning techniques such as GANs[4, 9, 14] or on extracting intrinsic feature information from existing data to facilitate adaptation to the target domain. In contrast, the model-driven

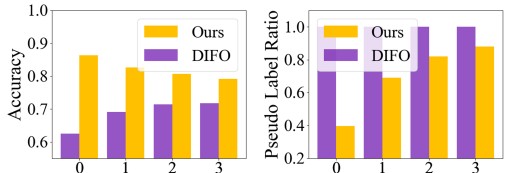

(a) Pseudo label accuracy  (b) Pseudo labeled sample ratio

Figure 1: Limitations of the existing state-of-the-art method[29] and comparison with the proposed approach on Clipart (target domain) of the Office-Home dataset. Comparison of (a) pseudo label accuracy and (b) pseudo label assignment ratio. The $x$-axis represents the training iterations.

approach[12, 1, 29] operates in a self-supervised learning paradigm, where supervision is derived from the target domain data based on the predictions (pseudo labels) generated by the pre-trained source model. It generally outperforms data-based approaches and has become the most widely adopted paradigm in SFDA. A representative method of the model-driven approach is SHOT[12], which generates pseudo labels for target domain samples based on feature space analysis. These pseudo labels are subsequently utilized for entropy minimization, enhancing the pre-trained source model's performance in the target domain. AdaContrast[1] performs batch-wise pseudo label generation and refinement for test-time adaptation. It identifies the nearest neighbors of each target sample in the feature space based on cosine similarity and assigns pseudo labels by averaging the target model's predicted probabilities. The pre-trained source model is then trained using a combination of a supervised loss calculated by using pseudo labels and contrastive learning. These approaches provide strong supervision in unlabeled settings, facilitating domain adaptation. However, the pre-trained source model (i.e., **target model**) exhibits instability in the target domain, which leads to unreliable pseudo labels in the early training stages and makes it difficult for the target model to establish a clear learning direction.

To mitigate this issue, DIFO[29] employs CLIP[23], a vision-language (ViL) model with strong zero-shot capabilities, to enhance the quality of pseudo labels. Additionally, it introduces a mutual information (MI)-based loss function that enables prompt learning in CLIP and knowledge distillation between CLIP and the target model. Despite achieving state-of-the-art performance on multiple benchmark datasets, DIFO still faces challenges in assigning accurate pseudo labels to target domain samples, particularly in the initial training stages. Moreover, while MI-based prompt learning facilitates CLIP's adaptation to the target domain, its effectiveness is inherently limited by the fixed vision encoder extracting low-quality image features that lack clear inter-class separability.

Fig. 1 compares the proposed method with DIFO in terms of pseudo label accuracy and the pseudo-labeled sample assignment ratio on the Office-Home[34] dataset. As shown in Fig. 1, DIFO suffers

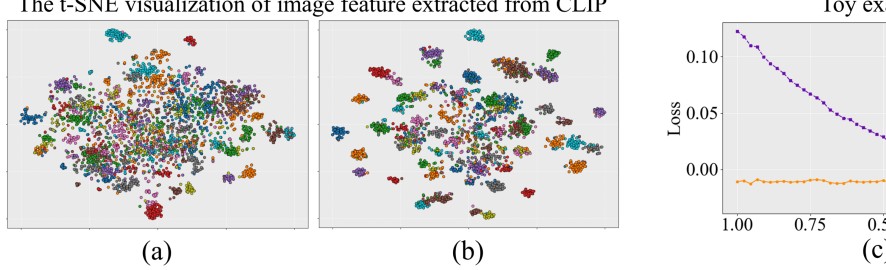

Figure 2: Limitations of CLIP utilized in the existing method DIFO[29]. (a) and (b) show the t-SNE visualization of image embeddings extracted by CLIP's vision encoder before and after the proposed vision encoder optimization, respectively. (c) represents comparison of Mutual information (MI) and Tsallis mutual information (TMI) under varying noise levels ($\sigma$). The $x$-axis represents the decreasing noise levels, while the $y$-axis denotes the computed loss.

from low pseudo label accuracy across the entire set of training samples. This issue hinders model convergence and introduces instability into the training process. Moreover, since CLIP is not pre-trained for specific tasks but rather exhibits task-agnostic properties, its predictions tend to reflect general and ambiguous patterns rather than task-specific distinctions. Therefore, a refinement process that accounts for this limitation is necessary. To this end, we propose two key strategies: (1) A learning framework that assigns pseudo labels only to samples whose predictions from the task-specific target adaptation model and the domain-invariant CLIP model agree. The remaining samples, which lack such agreement, are optimized using consistency-driven representation enhancement until sufficient feature quality is achieved. (2) A Tsallis mutual information (TMI)-based adaptation uncertainty-aware vision optimization strategy that dynamically adjusts training intensity based on the model's current adaptation uncertainty level, enabling robust adaptation even under large domain gaps.

In the first strategy, we refine the pseudo label generation process by assigning hard pseudo labels only to samples where the task-specific target adaptation model and domain-invariant CLIP agree in their predictions. This selective assignment allows us to extract highly reliable pseudo labels, combining task relevance and domain generalizability. Meanwhile, instead of assigning noisy labels to the remaining samples, we apply consistency training to boost the feature representation ability of the target model. This approach enables effective use of unlabeled data while minimizing label noise, thus improving overall adaptation performance.

In the second strategy, unlike DIFO—which adapts CLIP to downstream tasks via text prompt tuning—we directly fine-tune the CLIP vision encoder. Existing prompt tuning[24, 35, 42] can be limited when the image features extracted by the frozen vision encoder exhibit low clustering quality due to a large domain gap, as illustrated in Fig. 2(a). In this case, the text embeddings are forced to adapt to low-quality image representations, limiting performance. This problem becomes even more severe in SFDA, where no source data is available and the domain gap is large. To overcome this issue, we propose an alternative strategy: a TMI-based adaptation uncertainty-aware optimization method that enables effective feature alignment between the pre-trained CLIP knowledge and the target domain, thereby mitigating the limitations of prompt tuning. Specifically, existing prompt tuning is guided by a MI-based loss function between the target model and CLIP predictions. While this encourages aligned predictions between the two models, the standard MI loss does not sufficiently reflect the model's current training state. We demonstrate this limitation with a simple toy example in Fig. 2(c). This toy example simulates the training process of a model optimized using MI loss. In particular, we measure the MI between two probability vectors: one is a randomly generated probability vector (considered as ground truth, $p$), and the other is a noisy version of this probability vector (considered as the model prediction, $\hat{p}$). The detailed settings are provided in Appendix A. As the noise is gradually reduced, we observe how MI changes between these two distributions. As observed in Fig. 2(c), MI struggles to capture the overall changes in the probability distribution as the noise level decreases, making flexible loss measurement challenging (i.e., MI shows little meaningful variation despite gradual noise reduction). To address this limitation, we propose an approach that adaptively considers the model's adaptation uncertainty during CLIP's vision optimization. Specifically, we introduce a TMI loss function based on Tsallis entropy [33], which leverages our uncertainty-based adaptation index (UAI) to dynamically control the entropy curvature depending on the model's adaptation uncertainty. This UAI is used to compute TMI, allowing the model to adjust the strength of training based on its current adaptation uncertainty level and guiding the optimization process in a stage-aware manner. As shown in Fig. 2(c), the UAI enables TMI to encourage the utilization of diverse samples during the early unstable training phase (exploration), and to shift toward a stable optimization strategy (exploitation) for SFDA scenarios with large domain gaps. Consequently, TMI provides an effective optimization strategy in SFDA scenarios where a significant domain gap exists, balancing exploration and exploitation to achieve robust domain adaptation. Our contributions are summarized as follows:

- We propose a **Dual-perspective pseudo labeling strategy** that integrates a task-specific perspective, which emphasizes class-discriminative semantics, and a domain-invariant perspective, which focuses on robust generalization, thereby enabling more reliable supervision and structurally enriched representation learning.

- We introduce a **Tsallis mutual information**-based vision optimization strategy that leverages an **Uncertainty-based adaptation index** to dynamically regulate the optimization process by separating it into exploration and exploitation stages depending on the model's adaptation uncertainty level.

- Our method achieves performance improvements over state-of-the-art methods on Office-Home (+1.6%), VisDA-C (+1.4%), and DomainNet (+2.9%) datasets.

## 2 Preliminary

**Consistency training**  Consistency training[31, 25, 36, 20] is a widely used technique in semi-supervised learning that ensures the model produces consistent predictions when different augmentations are applied to the same input sample. This technique is particularly effective in scenarios with limited labeled data, as it provides a strong regularization effect, thereby improving model performance. In source-free domain adaptation (SFDA), where adaptation must be performed on an unlabeled target domain without access to labeled source domain data, the reliance on pseudo labels can introduce instability during training. By progressively aligning the model's predictions on augmented target samples, this approach mitigates the impact of pseudo label noise and enhances the model's ability to generalize to the target distribution.

**Tsallis entropy**  The conventional Shannon entropy-based model training process can lead to over-confidence in the presence of label noise[41, 17]. Tsallis entropy[33] offers a broader representational capacity in entropy computation compared to Shannon entropy, providing a potential solution to this issue. Tsallis entropy is defined as follows:

$$\ln_q(x) = \frac{x^{1-q} - 1}{1 - q}, S_q(p_i) = -\sum_i p_i \ln_q(p_i). \tag{1}$$

Here, $\ln_q(\cdot)$ and $S_q(\cdot)$ denote Tsallis logarithm and Tsallis entropy, respectively. $p_i$ represents the probability of event $i$ in a given discrete probability distribution. $q$ is known as the entropy index, controlling the degree of non-extensivity in the entropy formulation. As the entropy index $q$ approaches one, Tsallis entropy asymptotically converges to Shannon entropy. If $q$ is greater than one, the variation in entropy values with respect to probability changes is smaller than in Shannon entropy. For an intuitive understanding of how different values of entropy index $q$ affect the shape of the entropy function, please refer to the visualization in Appendix B. From a model training perspective, a Tsallis entropy-based loss function with $q$ larger than one reduces the model's sensitivity to high-confidence predictions and promotes exploration of uncertain samples. Conversely, when $q$ is less than one, the entropy values change more sharply with the probability variations, leading to increase sensitivity to confident predictions, encouraging exploitation. This characteristic highlights that a loss function utilizing Tsallis entropy can effectively regulate the model's learning pace. By adjusting the entropy index $q$, the model can emphasize exploratory learning under high uncertainty and shift toward stable optimization (exploitation) as confidence increases.

## 3 Methodology

**Overview**  As shown in Fig. 3, the proposed method is built upon three core components: 1) Calibrated pseudo label generation (CPG), which generates two types of pseudo labels (soft and hard) for target domain samples whose predictions from the target model and CLIP agree. 2) Dynamic entropy-guided vision optimization (DVO), in which the soft pseudo labels produced by CPG are used within the proposed Tsallis mutual information (TMI) to optimize CLIP's vision encoder for the target domain. 3) Pseudo label matching (PLMatch) framework, which trains the target model by incorporating hard pseudo labels.

These core components are then devided into two hierarchical processes, a cycle-level process and an iteration-level process in consideration of computational cost and training stability. In the cycle-level process, components with high computational cost, such as CPG and DVO are executed, enabling the generation of reliable pseudo labels and the adaptation of CLIP's vision encoder. In contrast, the iteration-level process consists of PLMatch framework, which repeatedly trains the target model with fixed hard pseudo labels from the current cycle. Specifically, our training framework is organized into multiple cycles, each consisting of several iteration-level processes. Each cycle first generates pseudo labels and optimizes the CLIP vision encoder, followed by several iteration-level processes that train the target model with the fixed hard pseudo labels.

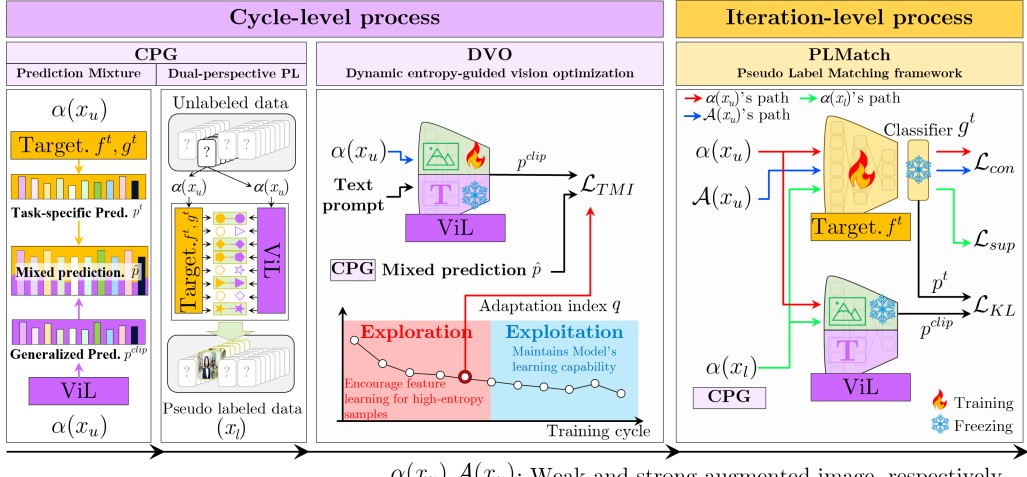

Figure 3: Overall structure of the proposed method.

## 3.1 Cycle-level process

### 3.1.1 Calibrated pseudo label generation (CPG)

In the cycle-level process, CPG generates pseudo labels for training both the CLIP vision encoder and the target model (pre-trained source model) by combining their prediction probabilities. Note that we obtain two types of pseudo label ($\hat{p}, \hat{y}$) in CPG, performing prediction mixture (soft label, $\hat{p}$) and dual-perspective pseudo labeling (hard label, $\hat{y}$). Soft pseudo label obtained from prediction mixture will be used in DVO for performing vision optimization of CLIP. Hard pseudo label acquired from dual-perspective pseudo labeling will be employed in PLMatch which is represented in Fig. 3 for training the target model.

**Prediction mixture**   To obtain the prediction mixture ($\hat{p}$) which will be utilized in DVO as a target label, the prediction probabilities for a target domain sample from CLIP ($p^{clip}$) and the target model ($p^t$) can be obtained as follows:

$$p_i^{clip} = \phi\left( \frac{sim(f^V(\alpha(x_i)), \theta^T)}{\tau} \right), p_i^t = \phi(g^t(f^t(\alpha(x_i)))), \tag{2}$$

where $f^V(\alpha(x_i))$ and $\theta^T$ ($\in \mathbb{R}^{(K \times D)}$) represent the weak augmented image feature which is the output of CLIP vision encoder $f^V(\cdot)$ and the text feature which is the output of CLIP's text encoder with a sentence in the format of "a photo of [class]"[23], respectively. $f^t(\cdot)$ and $g^t(\cdot)$ are the feature encoder and classifier of the target model, respectively. $\phi(\cdot)$ and $sim(\cdot, \cdot)$ denote the softmax function and the cosine similarity function, respectively. $\tau$ is a pre-trained temperature parameter[23]. $p_i^{clip}$ and $p_i^t$ denote the prediction vector ($\in \mathbb{R}^K$) for the $i$-th weakly augmented target sample $\alpha(x_i)$ from CLIP and target model, respectively. $K$ is the number of classes and $D$ is the latent dimension. Subsequently, the mixed prediction for a target domain sample is obtained by averaging the predictions as follows:

$$\hat{p}_i = (p_i^{clip} + p_i^t)/2, \tag{3}$$

where $\hat{p}_i \in \mathbb{R}^K$ represents a soft pseudo label. Note that the soft pseudo label $\hat{p}_i$ is utilized in DVO, which is the cycle-level process.

**Dual-perspective pseudo labeling**   To ensure high-confidence pseudo labels and mitigate the risk of noisy supervision, we employ a dual-perspective pseudo labeling strategy that selectively assigns pseudo labels only to samples where the predictions of CLIP ($p^{clip}$) and the target model ($p^t$) are aligned. Formally, we implement this selection mechanism using a binary mask $m_i$, where samples meeting the alignment criteria are assigned to pseudo labels, while others remain unlabeled:

$$\hat{y}_i = \delta(\hat{p}_i), m_i = \mathbf{1}\left[\operatorname*{argmax}_c(p_i^{clip}) = \operatorname*{argmax}_c(p_i^t)\right]. \tag{4}$$

Here, $\delta(\cdot)$ and $\mathbf{1}[\cdot]$ are a one-hot encoding function and indicator function, respectively. $m_i = 1$ if the predictions of both models are identical, and $m_i = 0$ otherwise. Note that the hard pseudo label $\hat{y}_i$ is utilized in PLMatch which is the iteration-level process.

### 3.1.2 Dynamic entropy-guided vision optimization (DVO)

**Uncertainty-based adaptation index** We define the uncertainty-based adaptation index $\psi$ to dynamically control the entropy behavior of TMI in response to the model's adaptation uncertainty. The $\psi$ allows balancing flexibility ($\psi > 1$) and convergence ($\psi < 1$) throughout training, and is directly incorporated into the computation of Tsallis mutual information. Specifically, likely with the entropy index $q$ in Eq. 1, when the model exhibits high adaptation uncertainty, the adaptation index $\psi$ is adjusted above one, resulting in a flatter curvature of the entropy function and encouraging learning from low-confidence samples (exploration process). Conversely, as the model's uncertainty decreases, the adaptation index $\psi$ is reduced below one, which sharpens the curvature of the entropy function and promotes stable learning centered on high-confidence samples (exploitation process). For this, to comprehensively reflect the model's uncertainty, the batch-wise normalized entropy $\tilde{H}(\tilde{p}^{clip})$ is firstly computed as follows:

$$\tilde{p}^{clip} = \frac{1}{N} \sum_{i=1}^{N} p_i^{clip}, \tilde{H}(\tilde{p}^{clip}) = \left( -\sum_{j=1}^{K} \tilde{p}_j^{clip} \log\left(\tilde{p}_j^{clip}\right) \right) / \log K, \tag{5}$$

where $\tilde{p}^{clip}$ denotes the averaged prediction vector of CLIP across a batch of size $N$. $\tilde{p}_j^{clip}$ denotes the $j$-th class probability of the averaged prediction vector $\tilde{p}^{clip}$, and $K$ is the number of classes. By quantifying the model's predictive uncertainty through this approach, we utilize EMA to dynamically adjust the adaptation index $\psi$. During the early training phase, when a higher learning rate is beneficial, $\psi$ is set to be greater than one to encourage exploration. As training progresses, the model requires more stable optimization, leading adaptation index $\psi$ to gradually decrease below one, with its variations being smoothly regulated:

$$\psi = \mu\psi + (1 - \mu)\tilde{H}(\tilde{p}^{clip}), \tag{6}$$

where $\mu$ acts as a momentum parameter that combines the previously computed $\psi$ with the current $\tilde{H}(\tilde{p}^{clip})$.

**Tsallis mutual information (TMI)** Subsequently, the standard MI loss[7] is replaced with Tsallis mutual information (TMI), where the adaptation index $\psi$ dynamically adjusts the entropy behavior based on the model's uncertainty. To optimize the CLIP vision encoder, the TMI $I_\psi(p^{clip}, \hat{p})$ between the CLIP prediction $p^{clip}$ and the soft pseudo label $\hat{p}$ is defined as follows:

$$\ln_\psi(x) = \frac{x^{1-\psi} - 1}{1 - \psi}, I_\psi(p^{clip}, \hat{p}) = \sum_{k=1}^{K} \sum_{k'=1}^{K} P_{kk'} \cdot \ln_\psi\left( \frac{P_{kk'}}{P_k P_{k'}} \right), \tag{7}$$

where $P_{kk'}$ denotes the joint probability of the $k$-th class probability from CLIP prediction $p^{clip}$, and the $k'$-th class from the soft pseudo label $\hat{p}$. $P_k$ and $P_{k'}$ are the marginal probabilities, representing the probability vectors of $p^{clip}$ and $\hat{p}$, respectively. For details on how the joint and marginal probabilities are computed for TMI, please refer to Appendix C.

**CLIP vision encoder optimization** Tsallis mutual information (negative term) is used as a loss function to optimize the CLIP vision encoder, encouraging the model to maximize the mutual information between the CLIP prediction ($p^{clip}$) and the soft pseudo label ($\hat{p}$). This is formulated as:

$$\mathcal{L}_{TMI} = -\min_{f_V} I_\psi(p^{clip}, \hat{p}). \tag{8}$$

By minimizing $\mathcal{L}_{TMI}$, the CLIP vision encoder is trained to enhance the alignment between the CLIP prediction $p^{clip}$ and the soft pseudo label $\hat{p}$, while simultaneously adapting the adaptation index $\psi$ to regulate the balance between exploration and exploitation during training.

## 3.2 Iteration-level process

### 3.2.1 Pseudo label matching framework (PLMatch)

PLMatch, which belongs to the iteration level process effectively (1) performs supervised learning using hard pseudo labels, (2) leverages unlabeled samples through consistency training, and (3) conducts CLIP-guided knowledge distillation.

**Pseudo supervision**   The target model is trained by minimizing the cross-entropy loss over the unlabeled set, using the hard pseudo label $\hat{y}_i$ and the mask $m_i$ obtained from dual-perspective pseudo labeling in CPG (Eq. 4). The corresponding formulation is as follows:

$$z_i^t = g^t\big(f^t(\alpha(x_i))\big), \quad p_i^t = \phi(z_i^t),$$

$$\mathcal{L}_{sup} = -\frac{1}{N_l}\sum_{i=1}^{N}\sum_{j=1}^{K}\hat{y}_{i,j}\log(p_{i,j}^t)\cdot m_i, \tag{9}$$

where $z_i^t$ is the logit output of the target model for $\alpha(x_i)$. $N_l$ is the number of pseudo labeled samples.

**Consistency training**   In addition to pseudo supervision, we apply consistency training to all target domain samples encouraging the target model to learn more robust feature representations rather than being overly confident in misclassifying samples that have not been assigned pseudo labels. As illustrated in Fig. 3, consistency training leverages both weakly augmented ($\alpha(x_i)$) and strongly augmented ($\mathcal{A}(x_i)$) versions of the same image ($x_i$). The training process is formulated as follows:

$$\mathcal{L}_{con} = -\frac{1}{N}\sum_{i=1}^{N}KL\big(p^t(\alpha(x_i))\big\|p^t(\mathcal{A}(x_i))\big), \tag{10}$$

where $p^t(\alpha(x_i))$ and $p^t(\mathcal{A}(x_i))$ represent the prediction vectors of $\alpha(x_i)$ and $\mathcal{A}(x_i)$ through the target model. $KL(\cdot\|\cdot)$ denotes the Kullback-Leibler (KL) divergence between the two input distributions.

**CLIP-guided knowledge distillation**   We further perform consistency training by minimizing the KL divergence between the prediction vectors from CLIP ($p^{clip}$) and the target model ($p^t$). The loss function for aligning the predictions of both models is formulated as follows:

$$\mathcal{L}_{KL} = -\frac{1}{N}\sum_{i=1}^{N}KL\big(p^t(\alpha(x_i))\big\|p^{clip}(\alpha(x_i))\big), \tag{11}$$

This alignment stabilizes the target model's feature representation, leading to more reliable predictions and improved generalization in the target domain.

**Target model training**   The three previously described loss functions are utilized for training the target model. These loss functions are combined with different weighting factors to formulate the final loss function, as defined below:

$$\mathcal{L}_{target} = \alpha\mathcal{L}_{sup} + \beta\mathcal{L}_{con} + \gamma\mathcal{L}_{KL}. \tag{12}$$

Here, $\alpha$, $\beta$, and $\gamma$ control the balance between losses, as further described in the experiments.

## 4   Experiments

**Datasets**   To evaluate the effectiveness of our proposed method, we conducted experiments on three widely used benchmark datasets for source-free domain adaptation: Office-Home[34], VisDA-C[22], and DomainNet-126[21]. These datasets encompass diverse domains and category scales. Detailed descriptions of each dataset are provided in Appendix D.1.

**Implementation details**   For reproducibility and clarity, detailed implementation settings—including backbone architectures, optimizer configurations, data augmentation strategies, and hyperparameters for each dataset—are thoroughly described in Appendix D.2.

Table 1: Accuracies (%) on Office-Home dataset for ResNet50-based methods and VisDA-C for ResNet101-based methods. "SF" denotes source-free methods, while domain notation are as follows: A (Art), C (Clipart), R (Real-world), P (Product), Sy (Synthesis), and Re (Real). Bold text highlights improvements over prior methods.

| Method | Venue | SF | A→C | A→P | A→R | C→A | C→P | C→R | P→A | P→C | P→R | R→A | R→C | R→P | Avg. | Sy → Re |
|---|---|---|---|---|---|---|---|---|---|---|---|---|---|---|---|---|
| Source | – | | 43.7 | 67.0 | 73.9 | 49.9 | 60.1 | 62.5 | 51.7 | 40.9 | 72.6 | 64.2 | 46.3 | 78.1 | 59.2 | 49.1 |
| SHOT | ICML20 | ✓ | 55.0 | 78.8 | 81.3 | 69.1 | 79.1 | 79.0 | 68.0 | 54.8 | 81.8 | 73.6 | 58.9 | 83.5 | 71.9 | 82.2 |
| NRC | NIPS21 | ✓ | 53.8 | 76.6 | 78.6 | 64.0 | 73.9 | 73.1 | 61.9 | 52.2 | 77.5 | 70.3 | 56.6 | 81.9 | 69.8 | 82.1 |
| GKD | IROS21 | ✓ | 56.6 | 78.3 | 82.2 | 69.4 | 80.5 | 78.6 | 67.2 | 55.3 | 82.5 | 74.3 | 59.7 | 84.1 | 72.4 | 82.6 |
| AdaCon | CVPR22 | ✓ | 47.2 | 75.1 | 75.5 | 60.7 | 73.3 | 73.2 | 60.2 | 45.2 | 76.6 | 65.6 | 48.3 | 79.1 | 65.0 | 86.8 |
| CoWA | ICML22 | ✓ | 56.4 | 79.2 | 80.9 | 68.6 | 78.6 | 75.8 | 68.0 | 55.9 | 82.5 | 69.0 | 65.8 | 79.9 | 68.6 | 84.3 |
| SCLM | CVPR23 | ✓ | 58.2 | 80.3 | 81.5 | 69.3 | 79.0 | 80.7 | 61.6 | 65.9 | 53.8 | 67.5 | 64.3 | 76.0 | 64.7 | - |
| TPDS | IJCV23 | ✓ | 59.3 | 80.3 | 82.1 | 70.6 | 79.4 | 80.9 | 69.8 | 56.8 | 82.1 | 74.5 | 61.2 | 85.3 | 73.5 | 87.6 |
| DIFO | CVPR24 | ✓ | 70.6 | 90.6 | 88.8 | 82.5 | 90.6 | 88.8 | 80.9 | 70.1 | 88.9 | 83.4 | 70.5 | 91.2 | 83.1 | 90.0 |
| Ours | – | ✓ | **73.6** | 90.4 | **91.0** | **83.6** | **90.7** | **90.9** | **82.7** | **73.7** | **91.2** | **83.6** | **74.0** | **91.2** | **84.7** | **91.4** |

Table 2: Accuracies (%) on DomainNet-126 for methods with ResNet50. Domain indentation are as follows: C(Clipart), P(Painting), R(Real), and S(Sketch).

| Method | Venue | SF | C→P | C→R | C→S | P→C | P→R | P→S | R→C | R→P | R→S | S→C | S→P | S→R | Avg. |
|---|---|---|---|---|---|---|---|---|---|---|---|---|---|---|---|
| Source | – | | 44.6 | 59.8 | 47.5 | 53.3 | 75.3 | 46.2 | 55.3 | 62.7 | 46.4 | 55.1 | 50.7 | 59.5 | 54.7 |
| SHOT | ICML20 | ✓ | 63.5 | 78.2 | 59.5 | 67.9 | 81.3 | 61.7 | 67.7 | 67.6 | 57.8 | 70.2 | 64.0 | 78.0 | 68.1 |
| NRC | NIPS21 | ✓ | 62.6 | 77.1 | 58.3 | 62.9 | 81.3 | 60.7 | 64.7 | 69.4 | 58.7 | 69.4 | 65.8 | 77.9 | 67.5 |
| GKD | IROS21 | ✓ | 61.4 | 77.4 | 60.3 | 69.6 | 81.4 | 63.2 | 68.3 | 68.4 | 59.5 | 71.5 | 65.2 | 77.6 | 68.7 |
| AdaCon | CVPR22 | ✓ | 60.8 | 74.8 | 55.9 | 62.2 | 78.3 | 58.2 | 63.1 | 68.1 | 55.6 | 67.1 | 66.0 | 75.4 | 65.4 |
| CoWA | ICML22 | ✓ | 64.6 | 80.6 | 60.6 | 66.2 | 79.8 | 60.8 | 69.0 | 67.2 | 60.0 | 69.0 | 65.8 | 79.9 | 68.6 |
| PLUE | CVPR23 | ✓ | 59.8 | 74.0 | 56.0 | 61.6 | 78.5 | 57.9 | 61.6 | 65.9 | 53.8 | 67.5 | 64.3 | 76.0 | 64.7 |
| TPDS | IJCV23 | ✓ | 62.9 | 77.1 | 59.8 | 65.6 | 79.0 | 61.5 | 66.4 | 67.0 | 58.2 | 68.6 | 64.3 | 75.3 | 67.1 |
| DIFO | CVPR24 | ✓ | 76.6 | 87.2 | 74.9 | 80.0 | 87.4 | 75.6 | 80.8 | 77.3 | 75.5 | 80.5 | 76.7 | 87.3 | 80.0 |
| Ours | – | ✓ | **80.1** | **89.6** | **79.0** | **82.4** | **89.8** | **79.2** | **82.8** | **80.6** | **78.8** | **83.0** | **80.3** | **89.6** | **82.9** |

Table 3: Unified evaluation of component configurations and loss weight ablation on Office-Home and VisDA-C datasets.

| Component | | | Weights | | | Office-Home | | | | | VisDA-C |
|---|---|---|---|---|---|---|---|---|---|---|---|
| $\mathcal{L}_{sup}$ | $\mathcal{L}_{con}$ | $\mathcal{L}_{KL}$ | $\alpha$ | $\beta$ | $\gamma$ | →A | →C | →P | →R | Avg. | Sy→Re |
| ✓ | | | 0.4 | 0.2 | 0.4 | 76.5 | 67.5 | 86.8 | 86.2 | 79.2 | 90.2 |
| ✓ | ✓ | | 0.4 | 0.2 | 0.4 | 77.0 | 68.3 | 87.2 | 86.4 | 79.7 | 90.4 |
| ✓ | | ✓ | 0.4 | 0.2 | 0.4 | 82.9 | 73.2 | 90.8 | 90.9 | 84.4 | 91.0 |
| ✓ | ✓ | ✓ | 0.4 | 0.2 | 0.4 | **83.3** | **73.8** | 90.8 | **91.0** | 84.7 | **91.4** |
| ✓ | ✓ | ✓ | 0.5 | 0.1 | 0.4 | 83.2 | 73.5 | 90.8 | 90.9 | 84.6 | 91.4 |
| ✓ | ✓ | ✓ | 0.5 | 0.2 | 0.3 | 82.7 | 73.4 | **90.9** | 90.9 | 84.5 | 91.4 |
| ✓ | ✓ | ✓ | 0.3 | 0.1 | 0.6 | 82.9 | 73.5 | 90.8 | 90.9 | 84.5 | 91.3 |
| ✓ | ✓ | ✓ | 0.4 | 0.1 | 0.5 | 83.2 | 73.4 | 90.8 | 91.0 | 84.6 | 91.4 |
| ✓ | ✓ | ✓ | 0.3 | 0.4 | 0.3 | 83.0 | 73.2 | 90.8 | 90.7 | 84.4 | 91.2 |
| Fine-tune w/ KL | | | 0.4 | 0.2 | 0.4 | 79.9 | 67.0 | 87.9 | 88.8 | 80.9 | 89.3 |
| Fine-tune w/ MI | | | 0.4 | 0.2 | 0.4 | 83.2 | 73.1 | 90.7 | 90.9 | 84.5 | 91.4 |
| Fine-tune w/ TMI | | | 0.4 | 0.2 | 0.4 | **83.3** | **73.8** | 90.8 | **91.0** | 84.7 | **91.4** |

## 4.1 Main results

As shown in Tabs. 1 and 2, our method consistently surpasses prior state-of-the-art approaches across all benchmarks, with improvements of 1.6% on Office-Home, 1.4% on VisDA-C, and 3.0% on DomainNet-126. The gains are especially notable in challenging domains such as Clipart and Sketch, where large domain gaps and stylistic differences hinder existing methods. These results highlight the robustness of our method in handling severe domain shifts and improving generalizable feature adaptation.

## 4.2 Contribution of investigated components

The ablation study presented in Tab. 3 evaluates the effectiveness of the proposed components in our framework, incorporating results on the Office-Home and VisDA-C datasets. First, when using only the pseudo supervision loss ($\mathcal{L}_{sup}$) with hard pseudo labels, the proposed model achieves 79.2% average accuracy on the Office-Home dataset, already surpassing existing source-free methods except DIFO[29]. On VisDA-C, our model achieves 90.2% accuracy, outperforming the state-of-the-art method DIFO by +0.2%. This strong performance of pseudo supervision highlights the superiority of

the proposed pseudo labeling strategy, and a more detailed analysis of our pseudo labeling strategy is provided in Appendix D.3.

Incorporating $\mathcal{L}_{con}$ and $\mathcal{L}_{KL}$ boosts accuracy by 5.5% (84.7%), highlighting their role in enhancing adaptation stability. Additionally, optimizing the CLIP vision encoder with Tsallis mutual information (TMI) results in the best overall performance, outperforming both KL-based and mutual information (MI)-based approaches. Notably, TMI-based vision optimization yields a larger performance gain in the Clipart domain ($\rightarrow C$), indicating its effectiveness in mitigating domain shift when adapting to visually diverse domains.

## 4.3 Variations depending on the loss weighting

We conducted an ablation study on the Office-Home and VisDA-C datasets to investigate the effect of weights $\alpha$, $\beta$, and $\gamma$ corresponding to $\mathcal{L}_{sup}$, $\mathcal{L}_{con}$, and $\mathcal{L}_{KL}$, respectively. As shown in Tab. 3, the optimal configuration ($\alpha = 0.4$, $\beta = 0.2$, $\gamma = 0.4$) achieved the best overall accuracy of 84.7% on Office-Home and 91.4% on VisDA-C. Notably, the model maintains robust performance despite variations in loss weight settings. This setting also showed strong performance on $\rightarrow R$ (91.0%), but slightly reduced accuracy was observed when $\beta$ was lowered or $\gamma$ was increased, indicating the importance of consistency regularization and balanced loss combination.

## 4.4 Analysis of Tsallis mutual information

Figure 4 analyzes the optimization dynamics of Tsallis mutual information (TMI) and standard mutual information (MI) losses during domain adaptation from Art to Clipart in the Office-Home dataset. The top plot shows the adaptation index $\psi$ decreasing over iterations, adaptively adjusting the sharpness of the TMI. The bottom plot compares TMI and MI losses before and after the adaptation index $\psi$ approaches one. The shaded regions in both plots divide the training process into two stages: the early stage encourages exploration, allowing the model to learn from diverse and uncertain samples, while the later stage favors exploitation, focusing on confident and stable predictions. As shown in Fig. 4, TMI effectively regulates loss oscillations, likely due to its entropy index-controlled structure. The loss patterns indicate that TMI promotes an effective exploration-exploitation balance, assigning greater importance to uncertain samples early in training. This structured learning improves model stability and makes DVO a more robust solution for source-free do-

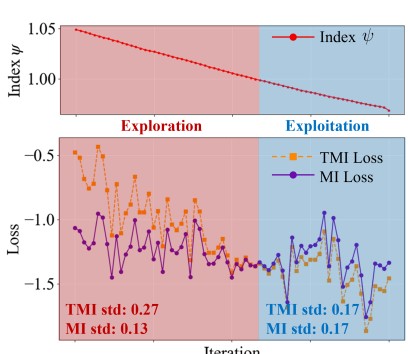

Figure 4: Visualization of the adaptation index $\psi$ (top) and the corresponding losses (bottom) for TMI and MI during adaptation from A→C in the Office-Home dataset.

main adaptation than standard MI-based training. The impact of TMI-based CLIP vision optimization on our framework is further analyzed in detail in AppendixD.3.

## 4.5 Analysis of Pseudo label matching framework

In all cases, PLMatch reduces the gap between confidence and classification accuracy more effectively than DIFO. Even in challenging domain shifts like R to A, PLMatch maintains relatively low expected calibration error (ECE)[19] (0.02) compared to DIFO (0.10). The superior calibration of PLMatch can be attributed to its dual-perspective pseudo labeling strategy, which prioritizes high-confidence samples initially and progressively refines pseudo labels over time. Additionally, consistency training enforces alignment between different augmented views, preventing the model from becoming overconfident in noisy pseudo labels. Additional results for other adaptation scenarios are provided in Appendix D.5.

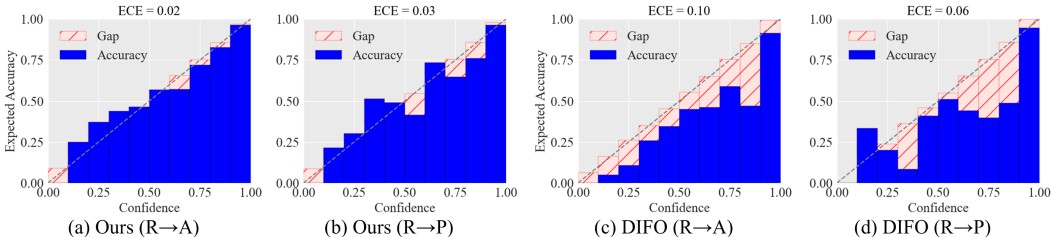

Figure 5: Reliability diagrams[19] comparing the proposed PLMatch and the existing state-of-the-art method, DIFO, on the Office-Home dataset. The source domain is Real-world (R), and the target domains are Art (A) and Product (P).

## 5  Conclusion and limitations

To address pseudo label noise in source-free domain adaptation (SFDA) and facilitate ViL model adaptation to the target domain, we introduce a novel pseudo labeling strategy that integrates task-specific perspectives and domain-invariant characteristics, along with an uncertainty-aware optimization strategy for the CLIP vision encoder. By leveraging domain-generalized CLIP predictions and task-specific model outputs, we selectively assign high-quality pseudo labels. Additionally, consistency training refines feature representations for samples awaiting pseudo labeling, enhancing robustness. Our uncertainty-based optimization approach balances exploration and exploitation, allowing the CLIP vision encoder to self-regulate its learning intensity. Empirical results demonstrate that our method outperforms prior approaches, particularly under challenging domain shifts with significant domain gaps. While our method requires optimizing the CLIP vision encoder, this introduces non-trivial computational overhead. Although we limit this process to fewer than eight update steps in our method, the added cost may still be a concern in resource-constrained settings, suggesting the need for lightweight adaptation strategies or parameter-efficient tuning methods in future work.

## 6  Acknowledgement

This research was supported by the National Research Foundation (NRF) funded by the Korean government (MSIT) under Grant RS2023-00208763, and RS-2025-02653611 (Intelligent Semiconductor Technology Development Program).

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

# Appendix

# Table of Contents

# A Implementation detail of toy example

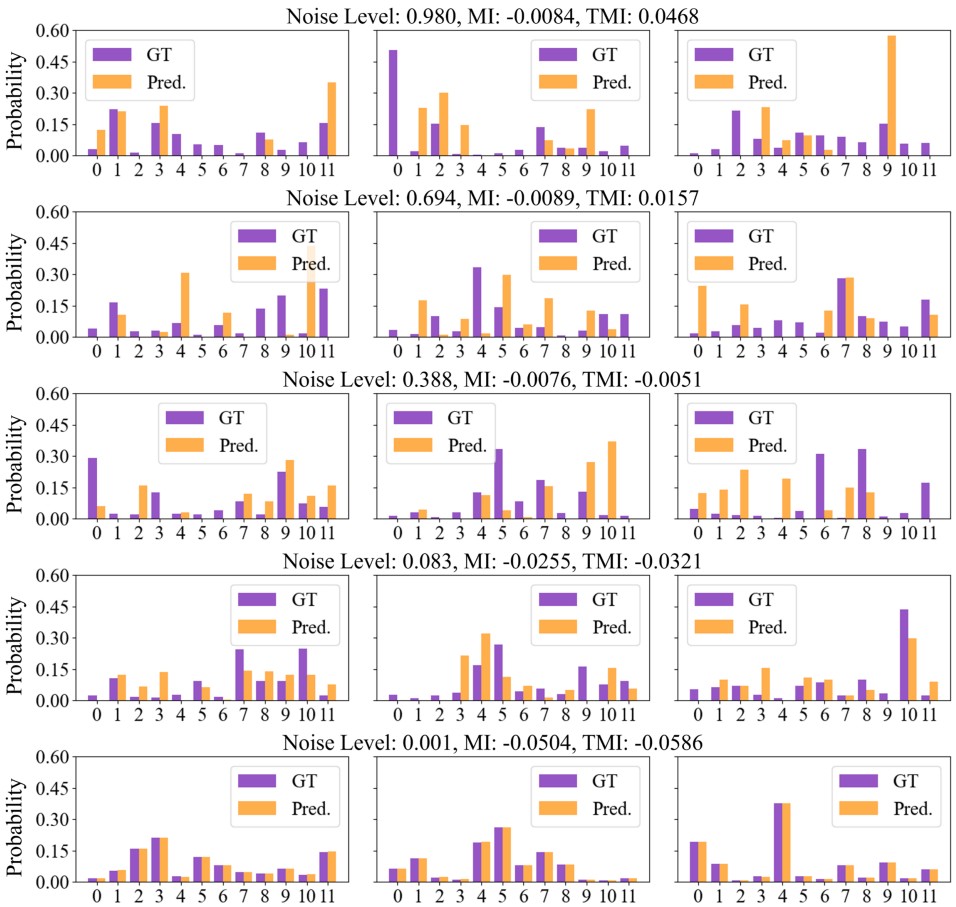

Figure 6: Visualization of class probabilities from a toy example designed to analyze mutual information (MI) sensitivity under different noise levels. The noise level refers to the weight applied to Gaussian noise with mean 0 and standard deviation 1, which is added to the soft pseudo label to simulate prediction. For each noise level, three sample predictions are shown. Each subplot compares a soft pseudo label (purple) against its noisy version (orange). As the noise level decreases from top to bottom, the noisy predictions become more similar to the pseudo labels. The $x$-axis represents the class index.

We further investigate the limitations of using standard mutual information (MI) as an optimization objective by designing a toy example to simulate the MI-based training process under decreasing uncertainty. As shown in Fig. 6, we generate soft pseudo labels by randomly sampling class probability vectors from a uniform distribution and then normalizing them via the softmax function to ensure they represent valid probability vectors. Controlled Gaussian noise is subsequently added to these vectors to simulate noisy predictions. For each noise level, three random samples are visualized to illustrate the distributional shift. Each subplot displays a pair of class probabilities: one representing the ground truth (pseudo label) and the other a noisy version (prediction) of it. We use the soft label as the ground truth to reflect the actual training setup, where mutual information (MI) is computed based on the soft pseudo label. As the noise level gradually decreases (from top to bottom), the noisy predictions become more concentrated and closely aligned with the pseudo label. However, as discussed in Fig. 2(c), despite the noticeable distributional improvements, the MI value remains relatively insensitive to these changes. This demonstrates that MI lacks the flexibility to dynamically capture information changes across varying prediction quality levels, particularly in uncertain early-stage adaptation scenarios. This observation supports the need for an alternative information measure like TMI, which dynamically modulates sensitivity through the proposed adaptation index $\psi$. By

doing so, TMI provides more meaningful optimization signals during the both high-uncertainty and stable training phases, leading to more robust adaptation performance.

## B Effect of entropy index $q$ on Tsallis entropy

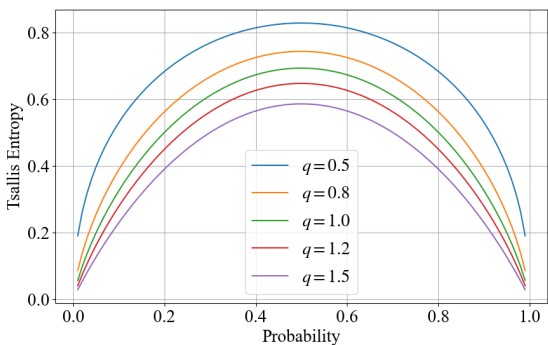

Figure 7: Effect of the entropy index $q$ on the shape of Tsallis entropy. Each curve represents the Tsallis entropy computed over probabilities ranging from 0 to 1, under different $q$ settings.

To better understand how the entropy index $q$ influences the behavior of Tsallis entropy[33], we visualize the entropy values over a range of probabilities in Fig. 7. The shape of the entropy function varies significantly with entropy index $q$. When $q < 1$, the entropy curve becomes sharper, increasing sensitivity to high-confidence predictions and encouraging the model to focus on confident samples (exploitation). Conversely, when $q > 1$, the curve becomes flatter, reducing the sensitivity to confidence and allowing the model to explore uncertain predictions more freely (exploration). This property allows Tsallis entropy to flexibly control the trade-off between exploration and exploitation during training by simply adjusting the entropy index $q$, which is especially beneficial in domain adaptation scenarios with high uncertainty.

## C Calculation of joint probabilities

To compute the Tsallis mutual information (TMI) between CLIP's prediction ($p^{clip}$) and the soft pseudo label ($\hat{p}$), we first construct a joint probability matrix $P$ following the standard formulation adopted in prior works[7]. This matrix $P_{kk'}$ captures how often each class prediction from CLIP co-occurs with each class prediction from the soft pseudo label, across all samples in a mini-batch. Specifically, for each sample in the batch, we calculate the outer product of the two probability vectors ($p^{clip}, \hat{p}$) and then take the average across all samples. This gives us a matrix $P_{kk'} \in \mathbb{R}^{K \times K}$, where $K$ and $N$ are the number of classes and samples, respectively. The calculation of matrix $P$ is followed:

$$P_{kk'} = \frac{1}{N} \sum_{i=1}^{N} P_{kk'}^{(i)} = \frac{1}{N} p_{clip}^{(i)} \cdot (\hat{p}^{(i)})^T, \tag{13}$$

The marginal probabilities $P_k$ and $P_{k'}$ are then obtained by summing this joint matrix $P_{kk'}$ across the rows and columns. The row sums represent the class probabilities from CLIP's prediction ($p^{clip}$), and the column sums represent those from the soft pseudo labels ($\hat{p}$). This is formulated as:

$$P_k = \sum_{k'} P_{kk'}, P_{k'} = \sum_{k} P_{kk'} \tag{14}$$

The joint probabilities and marginal probabilities are then jointly used in the computation of TMI, as formulated in Eq. 7.

## D  Experimental details

### D.1  Benchmarks

To evaluate the source-free domain adaptation performance of our proposed method, we used three benchmark datasets: Office-Home[34], VisDA-C[22], and DomainNet-126[21]. Office-Home comprises 15,500 images across four domains (Art, Clipart, Product, and Real-world), each with 65 categories. VisDA-C is a large-scale simulation-to-real dataset with over 152k images spanning 12 categories, where training images are synthetically generated, and validation images are real-world samples from MS-COCO. DomainNet-126 is a large-scale dataset derived from DomainNet. As a refined subset, DomainNet-126 contains 145k images from 126 classes, samples from four domains (Clipart, Painting, Real, and Sketch).

### D.2  Implementation details

Table 4: Hyper-parameter settings of the proposed method. $\eta_t$ and $\eta_V$ represent the learning rates of the target model and the CLIP vision encoder, respectively.

| Hyper-parameters | Office-Home | VisDA-C | DomainNet-126 |
|---|---|---|---|
| Batch size | 64 | 64 | 64 |
| Optimizer | SGD | SGD | SGD |
| Momentum | 0.9 | 0.9 | 0.9 |
| Weight-decay | 0.001 | 0.001 | 0.001 |
| $\eta_t$ | 0.01 | 0.001 | 0.001 |
| Cycle | 4 | 8 | 4 |
| Iteration | 4 | 4 | 4 |
| Adaptation index $\psi$ | 1.05 | 1.05 | 1.1 |
| Momentum $\mu$ | 0.99 | 0.999 | 0.99 |
| $\eta_V$ | 1e-7 | 1e-7 | 1e-7 |

We employed ResNet50[16] as the feature encoder for both the pre-trained source model and the target model on the Office-Home and DomainNet-126 datasets, while ResNet101[37] was used for VisDA-C. The target model was trained using stochastic gradient descent (SGD) as the optimizer with a momentum of 0.9 and a weight decay of 0.001. The learning rate was set to 0.01 for Office-Home and 0.001 for both VisDA-C and DomainNet-126. In contrast, CLIP's vision encoder (ViT/B-32[5]) was optimized using the Adam optimizer with a fixed learning rate of $1 \times 10^{-7}$. For consistency training, weak augmentation involves random cropping and horizontal flipping, while strong augmentation included randomly sized cropping, Gaussian blurring, and horizontal flipping to introduce greater variability. In terms of training schedule, the cycle was set to 4 for Office-Home and DomainNet-126, and 8 for VisDA-C, with each cycle comprising 4 iterations. The entropy index $q$ and momentum $\mu$ were adjusted slightly for each dataset to stabilize training. The weights $\alpha$, $\beta$, and $\gamma$ of loss functions were set to 0.4, 0.2, and 0.4 for all datasets, respectively. All experiments were conducted on an NVIDIA GeForce RTX 3090 GPU.

### D.3  Ablation details

#### D.3.1  Comparison between Tsallis mutual information (TMI) and mutual information (MI)

To examine the effectiveness and robustness of the proposed Tsallis mutual information (TMI), we conducted a comparison between mutual information (MI) and TMI across different random seeds, and additionally evaluated performance when replacing the backbone of the CLIP vision encoder. As shown in Tab. 5, the performance advantage of TMI over MI consistently appeared. Notably, the performance gap was more pronounced when using lightweight backbones such as ResNet[16, 37] variants, compared to the ViT-B/32[5] used in the final model.

Moreover, CLIP in our framework contributed in two major ways: knowledge distillation and pseudo label refinement. To assess TMI's contribution in these areas, we measured Expected Calibration Error (ECE) of CLIP, CLIP accuracy, and pseudo label accuracy on tasks where the performance improvement was most pronounced. As shown in Tab. 6, the results suggested that CLIP trained with TMI achieved a more stable and well-calibrated learning state than with MI, which positively affected the quality of the pseudo labels. These findings indicate that TMI-based training can be especially useful in various tasks.

Table 5: Comparison of mutual information (MI) and Tsallis mutual information (TMI) on the Office-Home dataset. For ViT-B/32, results are averaged over multiple random seeds, while RN50 and RN101 rows show performance when replacing the CLIP vision encoder backbone.

| Backbone | Method | A→C | A→P | A→R | C→A | C→P | C→R | P→A | P→C | P→R | R→A | R→C | R→P | Avg. |
|---|---|---|---|---|---|---|---|---|---|---|---|---|---|---|
| ViT-B/32 | Ours (w/MI) | 73.07 (±0.15) | **90.47** (±0.21) | 90.87 (±0.10) | 83.27 (±0.06) | 87.12 (±0.17) | **90.93** (±0.12) | 82.70 (±0.03) | 90.73 (±0.12) | 91.00 (±0.17) | 83.70 (±0.32) | 73.17 (±0.32) | 91.13 (±0.06) | 84.47 (±0.12) |
| | Ours (w/TMI) | **73.27** (±0.31) | 90.43 (±0.06) | **90.87** (±0.12) | **83.43** (±0.15) | **90.73** (±0.06) | 90.83 (±0.26) | **83.00** (±0.26) | 90.73 (±0.35) | **91.20** (±0.15) | **83.73** (±0.25) | **73.77** (±0.25) | **91.20** (±0.10) | **84.63** (±0.08) |
| RN50 | Ours (w/MI) | 59.0 | 84.8 | 84.6 | 78.7 | 85.5 | 86.8 | 78.7 | 60.3 | 87.2 | 78.7 | 60.4 | 85.2 | 77.6 |
| | Ours (w/TMI) | **60.5** | **84.9** | **86.9** | 78.5 | **85.7** | 86.7 | **79.0** | **61.6** | 86.9 | 78.7 | **61.5** | 85.2 | **78.0** |
| RN101 | Ours (w/MI) | 63.9 | 89.1 | 88.1 | 82.1 | 88.5 | 82.5 | 64.6 | 89.1 | 89.0 | 82.7 | 63.0 | 89.0 | 81.0 |
| | Ours (w/TMI) | **64.2** | **89.3** | **88.8** | **82.8** | **89.7** | **88.8** | **81.7** | **65.2** | 89.0 | 82.7 | **65.3** | **89.7** | **81.4** |

Table 6: Comparison of MI and TMI in Dynamic entropy-guided vision optimization (DVO) across different tasks. (ECE: Expected Calibration Error, CLIP Acc.: CLIP accuracy, PL Acc.: Pseudo label accuracy.)

| Task | Loss in DVO | ECE(↓) | CLIP Acc.(↑) | PL Acc.(↑) |
|---|---|---|---|---|
| A→C | MI | 0.20 | 77.0 | 71.8 |
| | TMI | **0.18** | **77.4** | **72.2** |
| P→C | MI | 0.20 | 76.9 | 71.6 |
| | TMI | **0.18** | **77.7** | **72.4** |
| R→C | MI | 0.19 | 77.6 | 71.6 |
| | TMI | **0.17** | **77.7** | **72.4** |

### D.3.2   Analysis on pseudo label

Table 7: Pseudo label accuracy(%) and its sampling ratio on Office-Home and DomainNet-126

| Dataset | Scenario | Cycle 1 | Cycle 2 | Cycle 3 | Cycle 4 |
|---|---|---|---|---|---|
| Office-Home | A→C | 87.05%(1722/4365) | 81.50%(3125/4365) | 79.04%(3679/4365) | 77.48%(3948/4365) |
| | A→P | 94.20%(2913/4439) | 92.62%(3997/4439) | 92.52%(4237/4439) | 92.03%(4315/4439) |
| | A→R | 95.74%(3101/4357) | 94.13%(3835/4357) | 93.35%(4075/4357) | 92.71%(4182/4357) |
| | C→A | 94.68%(1090/2427) | 90.79%(1878/2427) | 88.66%(2107/2427) | 87.57%(2213/2427) |
| | C→P | 94.75%(2631/4439) | 93.44%(3978/4439) | 92.46%(4228/4439) | 92.17%(4306/4439) |
| | C→R | 95.72%(2665/4357) | 94.08%(3834/4357) | 93.29%(4070/4357) | 92.49%(4196/4357) |
| | P→A | 93.47%(1149/2427) | 90.55%(1862/2427) | 88.89%(2089/2427) | 87.29%(2203/2427) |
| | P→C | 87.85%(1605/4365) | 81.56%(3118/4365) | 79.44%(3667/4365) | 78.22%(3930/4365) |
| | P→R | 95.85%(3014/4357) | 94.02%(3877/4357) | 93.27%(4084/4357) | 92.77%(4190/4357) |
| | R→A | 93.58%(1449/2427) | 90.71%(1926/2427) | 88.91%(2119/2427) | 87.62%(2213/2427) |
| | R→C | 84.60%(1916/4365) | 80.92%(3191/4365) | 78.90%(3721/4365) | 77.91%(3965/4365) |
| | R→P | 94.44%(3382/4439) | 92.95%(4058/4439) | 92.54%(3382/4439) | 92.18%(3382/4439) |
| | Avg. | 92.7 | 89.8 | 88.4 | 87.5 |
| DomainNet-126 | C→P | 92.97% (12638/30042) | 90.10% (22620/30042) | 88.64% (24935/30042) | 87.43% (26057/30042) |
| | C→R | 96.31% (40130/69622) | 94.26% (60876/69622) | 93.26% (64080/69622) | 92.40% (65707/69622) |
| | C→S | 92.76% (10306/24147) | 89.05% (17734/24147) | 87.48% (19818/24147) | 85.97% (20930/24147) |
| | P→C | 93.01% (9258/18523) | 90.04% (14455/18523) | 88.36% (15791/18523) | 87.33% (16530/18523) |
| | P→R | 96.16% (50654/69622) | 94.19% (61801/69622) | 93.25% (64571/69622) | 92.36% (66034/69622) |
| | P→S | 93.70% (10442/24147) | 89.30% (17924/24147) | 87.38% (19880/24147) | 85.90% (21002/24147) |
| | R→C | 92.88% (9623/18523) | 89.87% (14566/18523) | 88.33% (15875/18523) | 87.18% (16569/18523) |
| | R→P | 93.18% (17683/30042) | 89.93% (23479/30042) | 88.30% (25538/30042) | 87.07% (26394/30042) |
| | R→S | 95.85% (10886/24147) | 94.02% (17928/24147) | 93.27% (19943/24147) | 92.77% (21015/24147) |
| | S→C | 90.80% (9235/18523) | 87.98% (14436/18523) | 86.48% (15797/18523) | 85.35% (16506/18523) |
| | S→P | 95.18% (14221/30042) | 90.68% (22815/30042) | 88.89% (24931/30042) | 87.53% (26072/30042) |
| | S→R | 96.07% (40713/69622) | 94.17% (60981/69622) | 91.03% (64177/69622) | 92.36% (65778/69622) |
| | Avg. | 94.1 | 91.1 | 89.6 | 88.6 |

We conducted a quantitative analysis of pseudo label accuracy and sampling ratio per cycle on the Office-Home dataset [34]. As shown in Tab. 7, % values indicate pseudo label accuracy, and values in parentheses show the sampling ratio. Pseudo label accuracy declined across cycles (92.7%, 89.8%, 88.4%, 87.5%) as more pseudo labels introduced inevitable noise. However, prior work[18, 40] shows that such noise can improve generalization capability. This trade-off between pseudo label accuracy and sampling ratio is analyzed in detail in the following section.

### D.3.3 Trade-off between accuracy and the number of pseudo labels

Table 8: Comparison of pseudo label accuracy(%) and sampling ratio across cycles for different confidence thresholding (CT) strategies and our proposed pseudo labeling method on the Office-Home dataset.

| Threshold | Scenario | Cycle 1 | Cycle 2 | Cycle 3 | Cycle 4 |
|---|---|---|---|---|---|
| 0.85 | P→C | 98.82% (0.03) | 97.19% (0.20) | 94.72% (0.37) | 93.07% (0.47) |
| | R→A | 99.18% (0.15) | 98.67% (0.34) | 98.09% (0.45) | 97.88% (0.52) |
| 0.95 | P→C | 100% (0.007) | 98.79% (0.09) | 97.41% (0.19) | 96.70% (0.28) |
| | R→A | 99.03% (0.04) | 99.54% (0.18) | 99.39% (0.27) | 99.49% (0.32) |
| 0.85 to 0.95 | P→C | 98.82% (0.04) | 98.32% (0.18) | 96.49% (0.29) | 96.11% (0.34) |
| | R→A | 99.18% (0.15) | 98.64% (0.30) | 98.67% (0.37) | 98.75% (0.40) |
| Ours | P→C | 87.85% (0.37) | 81.56% (0.71) | 79.44% (0.84) | 78.22% (0.90) |
| | R→A | 93.58% (0.60) | 90.71% (0.79) | 88.91% (0.87) | 87.62% (0.91) |

Table 9: Comparison of target model classification performance between different CT strategies and our proposed method on the Office-Home dataset.

| Threshold | A→C | A→P | A→R | C→A | C→P | C→R | P→A | P→C | P→R | R→A | R→C | R→P | Avg. |
|---|---|---|---|---|---|---|---|---|---|---|---|---|---|
| 0.85 | 73.1 | 90.3 | 90.6 | 82.9 | 90.5 | 90.8 | 82.4 | 72.8 | 91.0 | **83.8** | 73.8 | 90.9 | 84.4 |
| 0.95 | 72.9 | **90.5** | 90.8 | 82.7 | 90.4 | 90.6 | 82.3 | 73.0 | 90.8 | 83.7 | 73.5 | 91.0 | 84.3 |
| 0.85 to 0.95 | 72.9 | 90.4 | 90.7 | 82.8 | 90.5 | 90.6 | 82.3 | 72.7 | 90.9 | 83.6 | 73.6 | 91.0 | 84.3 |
| Ours | **73.6** | 90.4 | 91.0 | **83.6** | **90.7** | **90.9** | 82.7 | 73.7 | **91.2** | 83.6 | **74.0** | **91.2** | **84.7** |

The drop in pseudo-label accuracy during later cycles is attributed to the increasing number of assigned pseudo labels, which highlights a natural trade-off between label quality and coverage. To further analyze this trade-off, we conducted additional experiments on the Office-Home dataset using confidence thresholding (CT) to assign pseudo labels, controlling the balance between pseudo label accuracy and quantity. As shown in Tab. 8, CT-based methods produced high-quality pseudo labels but labeled fewer samples than ours. In contrast, our approach achieved better overall performance and more stable training shown in Tab. 9. This aligns with the training strategy[2] of gradually incorporating lower confidence labels. This enables broader learning and mitigates overfitting, suggesting that our method ensures practical and stable label quality without complex thresholds or strategies.

### D.4 Effectiveness of CLIP for target model training

Table 10: Comparison of CLIP and the final target model on the Office-Home dataset. Each row reports the performance of the model indicated in the first Model column.

| Model | Setting | A→C | A→P | A→R | C→A | C→P | C→R | P→A | P→C | P→R | R→A | R→C | R→P | Avg. |
|---|---|---|---|---|---|---|---|---|---|---|---|---|---|---|
| CLIP | w/o tuning | 60.1 | 84.2 | 85.8 | 75.2 | 84.2 | 85.8 | 75.2 | 60.1 | 85.8 | 75.2 | 60.1 | 84.2 | 76.3 |
| | w tuning | 72.2 | 89.8 | 90.1 | 82.2 | 89.9 | 90.1 | 81.6 | 72.4 | 90.3 | 82.2 | 72.4 | 90.0 | 83.6 |
| Target | w/o CLIP-tuning | 65.8 | 87.9 | 89.2 | 80.7 | 88.1 | 89.1 | 80.4 | 66.4 | 89.0 | 81.1 | 66.4 | 88.3 | 81.0 |
| | w/ CLIP-tuning | **73.6** | 90.4 | **91.0** | **83.6** | **90.7** | **90.9** | **82.7** | **73.7** | **91.2** | **83.6** | **74.0** | **91.2** | **84.7** |

The primary objective of the proposed framework is to train the target model, not to directly use CLIP as the final model. The purpose of CLIP fine-tuning is therefore not to replace the target model, but rather to help the target model training with pseudo label refinement and knowledge distillation.

As shown in Tab. 10, the final target model (Proposed) outperformed both the frozen (CLIP w/o tuning) and fine-tuned CLIP models (CLIP w/ tuning), confirming that CLIP is best used as an auxiliary module rather than the main predictor.

### D.5 Expanded calibration analysis

As shown in Fig. 8, PLMatch consistently achieved a lower expected calibration error (ECE)[19] compared to DIFO[29] across various domain adaptation scenarios, indicating better alignment between confidence and accuracy.

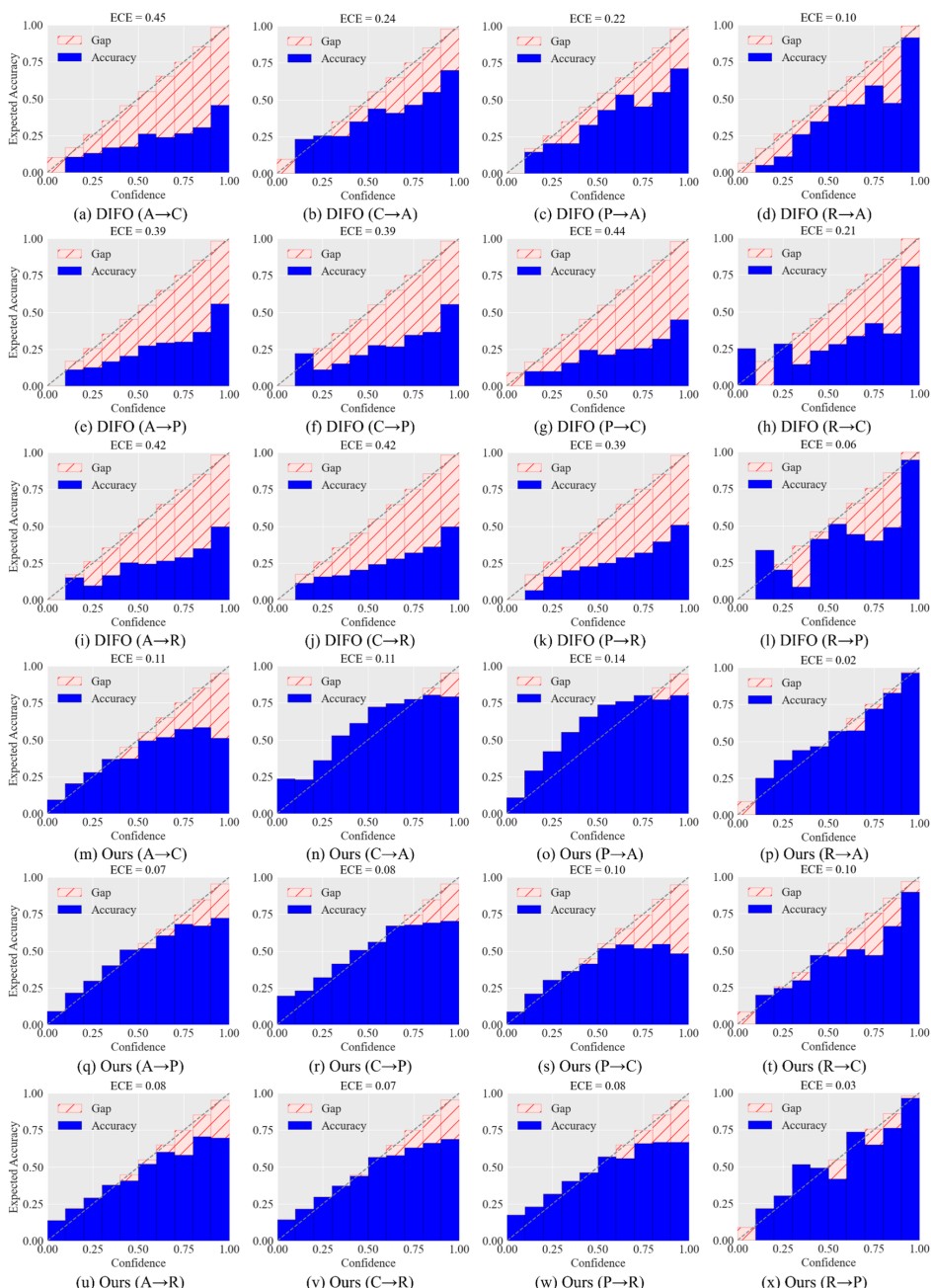

Figure 8: Calibration plots comparing DIFO and PLMatch across various domain adaptation scenarios on the Office-Home dataset. Each plot illustrates the expected accuracy versus confidence, where the gap represents the miscalibration.

## D.6 Feature distribution analysis

Figure 9 represents the t-SNE visualizations of the target domain features produced by various adaptation methods. Compared to existing approaches such as SHOT[12], GKD[28], NRC[38], and DIFO[29], our method forms more compact and well-separated clusters with clearer boundaries.

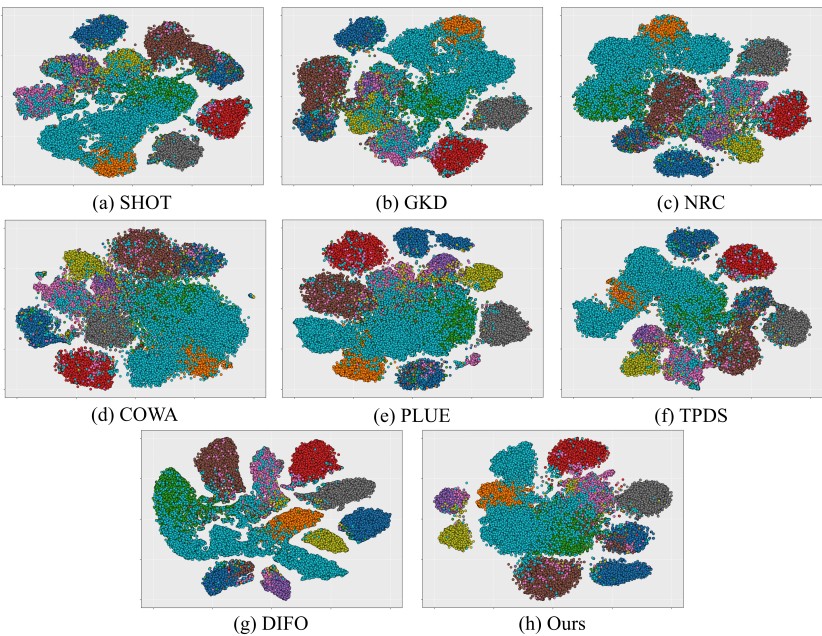

Figure 9: The t-SNE visualization of target domain features extracted by different domain adaptation methods on the VisDA-C dataset.

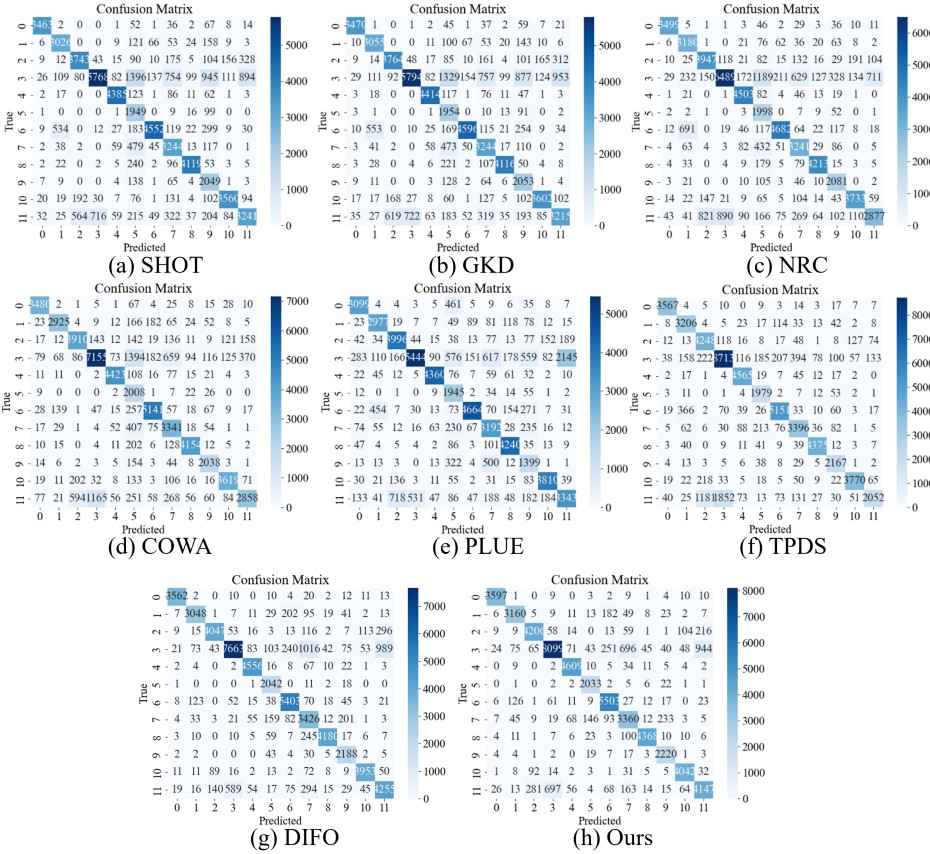

Figure 10: Comparison of class-wise prediction results across different SFDA methods on the VisDA-C dataset. Each matrix visualizes the classification accuracy between predicted (columns) and true (rows) labels. Darker diagonal lines indicate higher classification accuracy.

## D.7 Confusion matrix comparison

Figure 10 presents the confusion matrices of various SFDA methods, including SHOT[12], GKD[28], NRC[38], COWA[11], PLUE[13], TPDS[27], DIFO[29], and our proposed method on the VisDA-C[22] dataset. Each matrix illustrates the classification performance across all 12 classes, where diagonal elements indicate correctly predicted samples. Our method shows a clearly dominant diagonal pattern with significantly fewer off-diagonal elements compared to other approaches, indicating stronger class-wise discrimination and fewer misclassifications. This indicates the superior feature alignment and inter-class discrimination.

## D.8 Computational overhead

Table 11: Comparison of Computational Cost among SFDA Methods

| Method | TTB ↓ (s) | IT ↓ (ms) | GPU memory ↓ (GB) |
|--------|-----------|-----------|-------------------|
| SHOT   | 0.3621    | 0.608     | 7.759             |
| DIFO   | 0.6576    | 0.608     | 8.340             |
| Ours   | 0.5238    | 0.608     | 14.452            |

To provide a clearer analysis of computational limitations, we report training time per batch (TTB), inference time (IT) per sample and GPU memory usage for the Art→Clipart task from the Office-Home dataset[34]. Although our method requires higher GPU memory due to using both CLIP and the target model, its hierarchical structure—with separate cycle- and iteration-level processes—enables faster training than DIFO[29], which lacks such decomposition. Inference time per sample is identical across methods, as it depends only on the target model.

