# OpenReview forum: "DUET: Dual-Perspective Pseudo Labeling and  Uncertainty-aware Exploration & Exploitation  Training for Source-Free Domain Adaptation"
_NeurIPS.cc/2025/Conference — NeurIPS 2025 poster_

### Official Review · Reviewer_EPpg · 2025-06-22

**Clarity:** 3
**Significance:** 3
**Originality:** 3
**Rating:** 4
**Confidence:** 5

**Summary:**

The paper tackles source-free domain adaptation (SFDA) by jointly (i) calibrating pseudo labels from a task-specific classifier and CLIP (“dual-facet” hard labels) and (ii) fine-tuning the CLIP vision encoder with a Tsallis-mutual-information (TMI) loss modulated by an uncertainty-based adaptation index ψ. The combination is trained in repeated cycles of Calibrated Pseudo-label Generation (CPG), Dynamic Vision Optimization (DVO) and Pseudo-Label Matching (PLMatch) (Fig. 3). DUET achieves state-of-the-art accuracy on Office-Home, VisDA-C and DomainNet-126, improving the best prior method DIFO by 1.4 – 2.9 pp on average.

**Questions:**

See Weaknesses.

**Ethical Concerns:**

["NO or VERY MINOR ethics concerns only"]

**Final Justification:**

After reviewing it, I have no further questions and would prefer to retain my current rating.

**Quality:**

3

**Strengths And Weaknesses:**

Strengths:
1. Well-motivated dual supervision: Requiring agreement between CLIP and the task model before assigning a hard label reduces noise and yields better-calibrated confidence.
2. Novel uncertainty-aware TMI objective: The ψ-controlled entropy index smoothly anneals from exploration (ψ > 1) to exploitation (ψ < 1), stabilising training and showing clearer loss dynamics than standard MI.
3. Thorough empirical study: Main results on three benchmarks, plus ablations on loss components/weights (Tab. 3) and analyses of calibration, t-SNE, and confusion matrices.

Weaknesses:
1. Limited statistical rigour – Results are single-run averages; no variance or significance tests are reported.
2. Accuracy improves but requires up to 8 cycles × 4 iterations of target-model updates and several vision-encoder updates.
3. CLIP dependence – Method assumes access to a large ViL foundation model.

---

> ### Author Rebuttal · Authors · 2025-07-31
>
> **(W1) Limited statistical rigour**
>
> We appreciate the reviewer’s comment on the importance of statistical rigour in evaluating experimental results.
>
> To address this concern, we have conducted additional experiments using three different random seeds (2020, 2023, and 2025) and reported both the mean and standard deviation for each adaptation task across all three datasets (Office-Home [32], VisDA-C [20], and DomainNet-126 [19], respectively). These results provide a more comprehensive and reliable evaluation of our method.
>
> To maintain clarity in the main body, we will include the full statistical results in the Appendix section of the revised manuscript.
>
> **Table Q.** Experimental results and statistical rigor on the Office-Home dataset under varying seeds.
>
> |Seed|A→C|A→P|A→R|C→A|C→P|C→R|P→A|P→C|P→R|R→A|R→C|R→P|Avg.|
> |-|-|-|-|-|-|-|-|-|-|-|-|-|-|
> | 2020 |73.6|90.4|91.0|83.6|90.7|90.9|82.7|73.7|91.2|83.6|74.0|91.2|84.7|
> | 2023 |73.4|90.4|90.9|83.4|90.8|90.8|83.0|73.6|91.0|83.8|74.2|91.2|84.7|
> |2025|73.0|90.4|90.8|83.3|90.8|90.8|83.1|73.3|90.9|83.9|73.8|91.3|84.6|
> |**–**|**73.27 ± 0.3**|**90.43 ± 0.1**|**90.87 ± 0.1**|**83.43 ± 0.2**|**90.73 ± 0.1**|**90.83 ± 0.1**|**83.00 ± 0.3**|**73.33 ± 0.4**|**90.93 ± 0.3**|**83.73 ± 0.2**|**73.77 ± 0.3**|**91.20 ± 0.1**|**84.63 ± 0.1**|
>
> **Table R.** Experimental results and statistical rigor on the VisDA-C dataset under varying seeds.
>
> |Seed|plane|bike|bus|car|horse|knife|mcycle|person|plant|sktbrd|train|truck|Avg.|
> |-|-|-|-|-|-|-|-|-|-|-|-|-|-|
> |2020|98.7|91.1|89.8|79.1|98.2|97.8|95.1|84.2|96.0|97.3|95.4|73.9|91.4|
> |2023|98.6|91.4|88.5|78.6|98.3|98.1|95.2|84.9|96.0|97.1|95.4|73.6|91.3|
> |2025|98.7|89.7|88.2|79.1|97.9|97.5|96.2|85.8|96.5|97.3|95.0|72.8|91.2|
> |**–**|**98.7 ± 0.0**|**90.7 ± 0.7**|**88.8 ± 0.7**|**78.9 ± 0.2**|**98.1 ± 0.2**|**97.8 ± 0.2**|**95.5 ± 0.5**|**85.0 ± 0.7**|**96.2 ± 0.2**|**97.2 ± 0.1**|**95.3 ± 0.2**|**73.4 ± 0.5**|**91.3 ± 0.08**|
>
> **Table S.** Experimental results and statistical rigor on the DomainNet-126 dataset under varying seeds.
>
> |Seed|C→P|C→R|C→S|P→C|P→R|P→S|R→C|R→P|R→S|S→C|S→P|S→R|Avg.|
> |-|-|-|-|-|-|-|-|-|-|-|-|-|-|
> |2020|79.9|89.5|79.0|82.8|89.7|79.3|83.0|80.7|78.9|83.5|80.3|89.5|83.0|
> |2023|80.3|89.6|79.1|82.6|89.7|78.7|82.8|80.7|79.0|83.0|80.5|89.6|83.0|
> |2025|80.1|89.6|78.9|83.1|89.8|78.9|82.9|80.7|79.0|83.1|80.5|89.6|83.0|
> |**-**|**80.1 ± 0.2**|**89.6 ± 0.1**|**79.0 ± 0.1**|**82.8 ± 0.3**|**89.7 ± 0.1**|**79.0 ± 0.3**|**82.9 ± 0.1**|**80.7 ± 0.0**|**79.0 ± 0.1**|**83.2 ± 0.3**|**80.4 ± 0.1**|**89.6 ± 0.1**|**83.0 ± 0.1**|
>
> ---
> **(W2) Computational Overhead**
>
> We appreciate the reviewer’s observation regarding the increased number of updates during training.
> We do acknowledge that there is a trade-off between accuracy and the cost of updating the CLIP model parameters.
> To further investigate this, we conducted additional experiments on the VisDA-C dataset [20], varying the number of cycles and iterations while keeping the total number of target model updates fixed (the number of cycles × the number of iterations). The results indicate that increasing the number of CLIP vision encoder updates (conducted by cycle-level process) leads to consistent improvements in accuracy, even with the same number of target model updates.
> Based on these findings, although the proposed method inherently requires more computational resources due to the involvement of both CLIP and the target model, we would like to emphasize that it provides a flexible structure that allows users to adjust the trade-off between computational cost and performance. Moreover, our final setup carefully balances accuracy and training efficiency across all experiments.
>
>
> **Table T.** Comparison of target model performance with varying the number of CLIP updates (cycles) under a fixed number of target model updates (the number of cycles × the number of iterations). The asterisk (*) denotes the cycle-iteration combination employed in the final performance.
> |Cycle|Iteration|plane|bike|bus|car|horse|knife|mcycle|person|plant|sktbrd|train|truck|Avg.|
> |-|-|-|-|-|-|-|-|-|-|-|-|-|-|-|
> |4|8|98.6|88.5|**91.5**|78.2|98.0|97.4|94.6|**85.5**|95.6|97.0|94.4|69.8|90.8|
> |8*|4*|98.6|91.2|89.8|**79.8**|**98.1**|97.8|**95.0**|84.7|96.2|**97.2**|**95.0**|73.1|91.4|
> |16|2|**98.7**|**92.0**|90.0|79.0|**98.1**|**98.1**|**95.0**|85.2|**96.8**|97.0|94.8|**73.3**|**91.5**|
>
> |Cycle|Iteration|Training time (s)|
> |-|-|-|
> |4|8|17426|
> |8*|4*|18641|
> |16|2|21933|
> ---
> **(W3) CLIP dependency**
>
> We acknowledge the concern regarding CLIP dependence. However, we respectfully ask the reviewer to consider that our proposed method does more than simply utilize CLIP—it is fundamentally designed to provide deeper insights into SFDA through a carefully crafted training strategy and framework architecture.
>
> First, the proposed Tsallis Mutual Information (TMI) offers an important insight that SFDA training should adopt an exploratory strategy in the early stages and shift toward exploitation in the later stages. Moreover, TMI allows this transition to occur automatically based on the model’s adaptation status to the target domain.
>
> Second, our hierarchical training structure—comprising the cycle-level and iteration-level processes—and the dual-facet pseudo labeling strategy enable robust performance even without CLIP, as demonstrated in our ablation study.
>
>
> **Table U.** Comparison of target model performance with SHOT and our proposed framework without CLIP on the Office-Home dataset.
>
> |Settings|Venue|A→C|A→P|A→R|C→A|C→P|C→R|P→A|P→C|P→R|R→A|R→C|R→P|Avg.|
> |-|-|-|-|-|-|-|-|-|-|-|-|-|-|-|
> |**SHOT** [11]|PMLR 2020|55.0|78.8|81.3|69.1|79.1|79.0|68.0|54.8|81.8|73.6|58.9|83.5|71.9|
> |**AaD** [a]|NIPS 2022|59.3|79.3|82.1|68.9|79.8|79.5|67.2|57.4|83.1|72.1|58.5|85.4|72.7|
> |**C-SFDA** [b]|CVPR 2023|60.3|80.2|82.9|69.3|80.1|78.8|67.3|58.1|83.4|73.6|**61.3**|86.3|73.5|
> |**I-SFDA** [c]|CVPR 2024|**60.7**|78.9|82.0|69.9|79.5|79.7|67.1|**58.8**|82.3|74.2|**61.3**|**86.4**|73.4|
> |**SHOT+DPC** [d]|CVPR 2024|59.2|79.8|82.6|68.9|79.7|79.5|68.6|56.5|82.9|73.9|61.2|85.4|73.2|
> |**Target w/o CLIP**|-|58.1|**81.9**|**83.6**|**70.7**|**81.1**|**80.8**|**69.5**|55.7|**85.1**|**74.3**|59.9|86.0|**73.9**|
>
> Specifically, to assess the effectiveness of our framework independently of the CLIP model, we compared the performance of the target model across various SFDA methods that don't use CLIP. For fair comparison, we replaced CLIP-based pseudo labels with those generated by SHOT [11] in our SFDA method. This setting corresponds to Target w/o CLIP in **Table U**. Notably, both SHOT [11] and Target w/o CLIP use the same pseudo labels (from SHOT), yet our framework achieves a +2.0% accuracy improvement over SHOT [11]. This demonstrates that our hierarchical learning design and dual-facet strategy are effective for SFDA. Furthermore, our method outperforms all other methods as well as SHOT+DPC [d], a recent SHOT-based approach. This highlights that our framework is highly effective for SFDA and serves as a competitive baseline.
> In summary, while CLIP is effective in our framework, the strong results of our SFDA framework achieved even with pseudo labels generated by SHOT (not CLIP) show that the core strength of our method lies in the framework itself.
>
> ---
> **References**
>
> [a] Attracting and Dispersing: A Simple Approach for Source-Free Domain Adaptation. Shiqi Yang et al. NeurIPS 2022.
>
> [b] C-SFDA: A Curriculum Learning Aided Self-Training Framework for Efficient Source-Free Domain Adaptation. Nazmul Karim et al. CVPR 2023.
>
> [c] Understanding and Improving Source-Free Domain Adaptation from a Theoretical Perspective. Yu Mitsuzumi et al. CVPR 2024.
>
> [d] Discriminative Pattern Calibration Mechanism for Source-Free Domain Adaptation. Haifeng Xia et al. CVPR 2024.

---

> > ### Comment · Reviewer_EPpg · 2025-08-04
> >
> > Thank you for your rebuttal. After reviewing it, I have no further questions and would prefer to retain my current rating.

---

> > > ### Author Response · Authors · 2025-08-05
> > >
> > > We sincerely thank the reviewer for taking the time to read our rebuttal and for considering our clarifications. We truly appreciate your thoughtful evaluation and constructive feedback throughout the review process. In the final version of the manuscript, we will integrate the expanded discussions and clarifications presented in our rebuttal. Thank you once again for your invaluable insights.

---

### Official Review · Reviewer_9Bc5 · 2025-06-24

**Clarity:** 2
**Significance:** 2
**Originality:** 2
**Rating:** 5
**Confidence:** 4

**Summary:**

This work proposes a new method for source-free domain adaptation, consisting of several components as follows. To improve the quality of pseudo-labels, the proposed dual-facet pseudo-labeling strategy assigns labels only to target samples on which both the target model and CLIP agree. The vision encoder of CLIP is trained with a novel loss based on Tsallis Mutual Information (TMI), which has desirable theoretical properties. Leveraging TMI enables the method to gradually shift from exploration to exploitation. Finally, pseudo-labeled target samples are used to supervise the training of the target model; unlabeled samples are utilized for representation learning via consistency training; and a KL-divergence-based loss is applied to distill knowledge from CLIP to the target model. Experimental results demonstrate that the proposed method achieves competitive performance.

**Questions:**

1. I suggest that the authors provide a more detailed introduction to the principles and essential elements of Tsallis Entropy (TE) and Tsallis Mutual Information (TMI), as these concepts are not widely familiar to most readers. Additionally, the motivation for incorporating TE and TMI into the source-free domain adaptation (SFDA) framework should be more clearly justified. After all, this is the core contribution of the work.

2. The authors are encouraged to improve the clarity and readability of the manuscript. Careful language polishing will help ensure the proposed ideas are effectively conveyed.

3. A specific question: What exactly do the terms “cycle-level process” and “iteration-level process” mean in the context of this paper? This phrasing is somewhat unclear and may require clarification or rewording for better comprehension.

**Ethical Concerns:**

["NO or VERY MINOR ethics concerns only"]

**Final Justification:**

I have no further questions, and I am raising the score to 5. I hope the authors will revise their paper accordingly.

**Limitations:**

Yes.

**Paper Formatting Concerns:**

NO Paper Formatting Concerns.

**Quality:**

3

**Strengths And Weaknesses:**

Strengths:
The main contribution of this work is the introduction of Tsallis Entropy (TE) and Tsallis Mutual Information (TMI) into the area of source-free domain adaptation, which provides a novel and potentially valuable perspective.

Weaknesses:
1. Apart from TE and TMI, other components of the proposed method are relatively conventional. Dividing target samples into easy and hard ones is intuitive and has been explored in prior works [1, 2]. Utilizing unlabeled samples through consistency training is a standard technique in self-supervised learning, particularly in contrastive learning frameworks [3]. Knowledge distillation is also widely adopted when leveraging pre-trained models such as CLIP [4]. Therefore, the novelty of these components—except the TMI-based loss—is limited. Moreover, the introduction and motivation for incorporating TE and TMI are not sufficiently detailed or justified.
2. The writing quality of the manuscript is subpar, which significantly affects readability and comprehension. There are multiple grammatical errors, even in the Abstract, which should be carefully avoided. For instance, the abstract sentence: “In a self-supervised manner, relying on pseudo labels on target domain samples facilitates the domain adaptation performance providing strong supervision.”
is grammatically incorrect and semantically ambiguous. I suggest rephrasing it for clarity and conciseness. One possible revision could be: “In a self-supervised setting, pseudo labels on target domain samples provide strong supervision, thereby improving domain adaptation performance.” Overall, the language quality is not adequate for publication and requires substantial improvement to enhance clarity and readability.

References:

[1] Yang, Jianfei, Xiangyu Peng, Kai Wang, Zheng Zhu, Jiashi Feng, Lihua Xie, and Yang You. "Divide to Adapt: Mitigating Confirmation Bias for Domain Adaptation of Black-Box Predictors." *The Eleventh International Conference on Learning Representations.*

[2] M. Xia et al., “A Separation and Alignment Framework for Black-Box Domain Adaptation,” *Proceedings of the AAAI Conference on Artificial Intelligence*, Mar. 2024, pp. 16005–16013. doi: 10.1609/aaai.v38i14.29532.

[3] K. He, H. Fan, Y. Wu, S. Xie, and R. Girshick, “Momentum Contrast for Unsupervised Visual Representation Learning,” CVPR 2020

[4] Liang, Jian, Dapeng Hu, Jiashi Feng, and Ran He. "Dine: Domain adaptation from single and multiple black-box predictors." In Proceedings of the IEEE/CVF conference on computer vision and pattern recognition, pp. 8003-8013. 2022.

---

> ### Author Rebuttal · Authors · 2025-07-31
>
> **(W1, Q1) Limitied novelty beyond TE and TMI**
>
> Thank you for your thoughtful feedback. We sincerely appreciate the reviewer’s careful attention to whether the contributions of our work are sufficiently distinct and clearly communicated.
>
> * **Tsallis entropy**
>
> The contribution of our study lies in associating Tsallis entropy—which can either emphasize or suppress prediction uncertainty—with the model's degree of adaptation to the target domain. This enables the model to adopt different strategies for sample exploration and exploitation depending on its own adaptation stage. This strategy has not been previously explored in the SFDA field.
>
> * **Dual-facet pseudo labeling strategy**
>
> Moreover, the proposed dual-facet pseudo labeling strategy offers a simple yet practical approach. By using the intersection of predictions from CLIP (which provides a domain-invariant perspective) and the target model (which offers a task-specific perspective) as pseudo labels, our method delivers strong pseudo supervision without relying on complex pseudo labeling mechanisms. As shown in lines 273–277 of the main paper, using only these pseudo labels still achieves performance that surpasses most existing SFDA methods. We would greatly appreciate it if this aspect could also be considered as a meaningful contribution.
>
> Now, we will strengthen the expressions in the Introduction and Preliminary sections based on the contributions related to Tsallis entropy and Tsallis Mutual Information.
> Specifically, as shown in Appendix B, TE allows us to assign varying importance  to different levels of prediction confidence by adjusting the entropy index $\mathcal{q}$.
>
> > When $\mathcal{q > 1}$, the loss function emphasizes low-confidence predictions, encouraging broader exploration.
>
> > When $\mathcal{q < 1}$, the loss function focuses on high-confidence samples, enabling focused exploitation.
>
> Our method leverages this property by incorporating it into the mutual information objective, enabling a dynamic exploration–exploitation strategy during CLIP vision optimization.
> In the early training stages, it suppresses overconfident predictions and encourages more cautious learning.
> In later stages, it shifts toward reinforcing reliable samples.
> This exploration–exploitation trade-off, modulated by model uncertainty, is particularly effective in scenarios with large domain gaps, where overfitting to early predictions can harm generalization.
> To better highlight this unique contribution, we will revise the Introduction to more clearly articulate the role and motivation for using TE and the adaptation index in our learning framework.
>
> ---
> **(W2, Q2) Clarity and readability of the manuscript**
>
> We sincerely apologize for not ensuring sufficient clarity and readability in the main paper. We will thoroughly review the main paper and make sure to revise any unclear or low-quality expressions.
>
> - **Abstract**
>
> 1. As pointed out in Weakness 2, we will revise lines 2–4 to “In a self-supervised setting, pseudo labels on target domain samples provide strong supervision, thereby improving domain adaptation performance.”
> 2. In lines 4–6, we will change “However, the critical problem” to “However, a critical problem”.
> 3. In lines 9–10, we will revise “target samples with agreement between a target model and CLIP” to “target samples on which the target model's predictions and CLIP's predictions agree”.
>
> - **Introduction**
>
> 1. In line 44, we will change “paradigm in the SFDA” to “paradigm in SFDA”.
> 2. In line 51, we will revise “using a combination of supervised loss” to “using a combination of a supervised loss ”.
>
> - **Preliminary**
>
> 1. In line 127, we will update the phrase “that enforces the model to produce consistent predictions” to “that ensures the model produces consistent predictions”.
>
> - **Methodology**
>
> 1. In line 234, we will revise “leverages unlabeled samples using consistency training,” to “leverages unlabeled target samples through consistency training”.
>
> - **Experiments**
>
> 1. In lines 273–274, we will revise “First, when using only the pseudo supervision loss with hard pseudo labels,” to “First, when only the pseudo supervision loss with hard pseudo labels is used.”
>
>
>
> ---
> **(Q3) Clarification of "cycle-level process" and "iteration-level process"**
>
> We apologize for any confusion caused by the complexity of our framework design. To improve clarity, we will revise the Overview section of the manuscript to more clearly explain the two-level structure.
> For clearer understanding, we provide a summary table of the operations and learnable modules at each process stage, along with the overall pseudocode of the proposed framework.
>
> |Process|Key Operations|Learnable Module|
> |-|-|-|
> |**Cycle-level**|Generate pseudo labels from CLIP and target model predictions (CPG)| (None) |
> | |Fine-tune CLIP encoder using Tsallis Mutual Information objective (DVO)|CLIP vision encoder|
> | **Iteration-level**|Train target model with fixed pseudo labels from current cycle (PLMatch)|Target model|
>
> 1. Cycle-level process
> * Refers to the outer training loop in our framework.
> * In each cycle:
> > * Pseudo labels are generated based on the predictions of CLIP and the target model. (Calibrated pseudo label generation, CPG)
> > * The CLIP vision encoder is fine-tuned using the proposed Tsallis Mutual Information (TMI) objective to adapt to the target domain. (Dynamic entropy-guided vision optimization, DVO)
>
> 2. Iteration-level process
> * Refers to the inner training loop within each cycle.
> * In each iteration:
> > * The target model is trained using the pseudo labels fixed for the current cycle. (Pseudo label matching framework, PLMatch)
> * Unlike the cycle-level process, the iteration-level process does not update pseudo labels or CLIP but focuses soley on optimizing the target model.
>
> ```python
> # Pseudo-code for proposed framework
>
> for cycle in range(num_cycles):  # Cycle-level process
>     # Step 1: Calibrated Pseudo Label Generation (CPG)
>     soft_pseudo_labels, hard_pseudo_labels = generate_calibrated_pseudo_labels(CLIP, target_model, target_data)
>
>     # Step 2: Dynamic Entropy-guided Vision Optimization (DVO)
>     CLIP = optimize_CLIP_with_TMI(CLIP, soft_pseudo_labels, target_data)
>
>     for iteration in range(num_iterations):  # Iteration-level process
>         # Step 3: Pseudo Label Matching (PLMatch)
>         target_model = train_with_PLMatch(target_model, hard_pseudo_labels, target_data, CLIP)
>
>         # No updates to CLIP or pseudo labels here — only optimize the target model
> return target_model
> ```

---

> ### Comment · Reviewer_9Bc5 · 2025-08-04
>
> Thank you for your response. I have carefully reviewed your replies to my initial concerns and questions, as well as your rebuttals to the comments from other reviewers. I now have a clearer understanding of your work.
>
> The proposed dual-facet pseudo-labeling strategy leverages the agreement between the predictions of CLIP and the target model to generate reliable pseudo-labels. While the approach is intuitive and straightforward, I do not consider it a particularly significant contribution. Moreover, the term "dual-facet" feels unnecessarily mystifying and somewhat misleading.
>
> Could you further elaborate on Tsallis Entropy (TE) and Tsallis Mutual Information (TMI)? Specifically, I would appreciate:
>
> (1) A brief tutorial-style explanation of TE and TMI; and
>
> (2) A clear discussion of how these concepts contribute to improving source-free domain adaptation.
>
> Additionally, since your work focuses on source-free domain adaptation, I find the use of the term "source-free adaptation" in the title potentially confusing. It could mislead readers into questioning whether the paper addresses the well-established source-free domain adaptation problem.
>
> Finally, I encourage the authors to incorporate the relevant clarifications from the rebuttal into the final version of the manuscript to further enhance its clarity and completeness. Thank you.

---

> > ### Author Response · Authors · 2025-08-05
> >
> > Thank you very much for your thoughtful follow-up and taking the time to revisit our rebuttal. As you suggested, we will integrate all relevant explanations into the final version of the manuscript to enhance both its clarity and completeness. Thank you again for your constructive feedback.

---

> > > ### Comment · Reviewer_9Bc5 · 2025-08-06
> > >
> > > I would appreciate it if you could respond to the follow-up questions I raised earlier.

---

> > > > ### Author Response · Authors · 2025-08-06
> > > >
> > > > Thank you for your constructive feedback. I will promptly prepare responses to the newly raised follow-up questions.

---

> > > > > ### Author Response · Authors · 2025-08-06
> > > > >
> > > > > **Concern about dual-facet pseudo labeling and misleading term**
> > > > >
> > > > > Thank you for your thorough review and valuable suggestions.
> > > > >
> > > > > Basing pseudo-label assignment on agreement between CLIP and the target model on the proposed method brings two key advantages.
> > > > >
> > > > > **First**, it unifies model’s perspectives: although CLIP and the target model—each trained on different datasets—often yield divergent predictions on the same target data, the samples on which both views agree provide far more reliable pseudo labels for model training [a].
> > > > >
> > > > > **Second**, it offers learning opportunities across a more diverse set of samples, thereby enriching the model’s representational capacity. In contrast, conventional methods such as confidence-based pseudo labeling that assign pseudo labels solely via a confidence threshold train only on samples the model is already confident about, which can foster overconfidence [b]. By allocating pseudo labels based on prediction agreement rather than a single-model confidence, dual-facet pseudo labeling itself delivers strong supervision while enabling pseudo labeling across a more diverse set of samples than existing methods (see lines 273–276).
> > > > >
> > > > > We kindly ask you to reconsider your evaluation of dual-facet pseudo labeling in light of the points discussed above and thank you for pointing out that the term “dual-facet” may appear unclear. To enhance readability and ensure clarity, we will replace “ facet” with the more intuitive term “perspective”. We appreciate your valuable suggestion.
> > > > >
> > > > > To keep it simple, we initially used the shorthand “source-free adaptation” after confirming its use in several prior works [c], [d]. However, we will update the title and all occurrences to “source-free domain adaption” for clarity. Thank you for your insightful suggestion.
> > > > >
> > > > > ---
> > > > >
> > > > > **References**
> > > > >
> > > > > [a] Deep Co-Training for Semi-Supervised Image Recognition. Siyuan Qiao et al. ECCV, 2018.
> > > > >
> > > > > [b] Dynamic Pseudo Labeling via Gradient Cutting for High-Low Entropy Exploration. Jae Hyeon Park et al. CVPR, 2025.
> > > > >
> > > > > [c] Source-Free Adaptation Diagnosis for Rotating Machinery. Jinyang Jiao et al. IEEE TII, 2022.
> > > > >
> > > > > [d] Summit: Source-Free Adaptation of Uni-Modal Models to Multi-Modal Targets. Cody Simons et al. ICCV, 2023.

---

> > > > > > ### Author Response · Authors · 2025-08-06
> > > > > >
> > > > > > **A brief tutorial-style explanation of Tsallis Entropy (TE) and Tsallis mutual information (TMI)**
> > > > > >
> > > > > > Tsallis Entropy (TE) allows you to adjust how much high- and low-probabilities in prediction vectors contribute to entropy calculation depending on the entropy index q. In the context of classification tasks, for example:
> > > > > >  - When $q < 1$, larger class–probability values contribute more heavily to the entropy calculation.
> > > > > >  - When $q > 1$, the difference in weighting between large and small class-probabilities for entropy calculation is reduced.
> > > > > >
> > > > > > You can see this effect summarized in the table below:
> > > > > >
> > > > > > | Class confidence |$q < 1$|$q = 1 (Shannon)$|$q > 1$|
> > > > > > |-|-|-|-|
> > > > > > | **High-Probability** |Contributes **more** ↑|Contributes **as-is**|Contributes **less** ↓|
> > > > > > | **Low-Probability**  |Contributes **less** ↓|Contributes **as-is**|Contributes **more** ↑|
> > > > > >
> > > > > > Our Tsallis mutual information (TMI) builds on this property of Tsallis entropy by linking it to model uncertainty through a proposed uncertainty-based adaptation index $\psi$, enabling the network to self-adjust its learning weighting over the course of training:
> > > > > >
> > > > > > 1. **Early Training Stage (Exploration)**
> > > > > >
> > > > > > >The uncertainty-based adaptation index $\psi>1$ is applied when computing mutual information, functioning similarly to the entropy index $q$ in Tsallis entropy. This reduces the weighting gap between high- and low-probability predictions—preserving their relative ranking but preventing the model from becoming over-confident too quickly.
> > > > > >
> > > > > > 2. **Later Training Stage (Exploitation)**
> > > > > >
> > > > > > >As the model’s uncertainty decreases, uncertainty-based adaptation index $\psi$ is set to smaller than 1. This sharpens the focus on high-probability predictions, allowing the model to fully exploit the distributions it has learned so far.
> > > > > >
> > > > > > We believe this explicit exploration–exploitation strategy is particularly well-suited to SFDA tasks, where label noise can otherwise cause severe model collapse. By dynamically adjusting uncertainty-based adaptation index $\psi$, we offer a novel insight into maintaining stability and performance under noisy, source-free conditions.
> > > > > >
> > > > > > ---
> > > > > >
> > > > > > **A clear discussion of how these concepts contribute to improving source-free domain adaptation**
> > > > > >
> > > > > > The exploration–exploitation strategy was inspired by reinforcement learning (RL), where an agent interacts with a complex environment, selects actions, and learns to maximize the rewards it receives. In RL, there is a risk that noisy or misleading rewards will reinforce suboptimal policies, so agents are encouraged to explore a wide range of policies in the early stages of training and then exploit the best-performing policies in later stages.
> > > > > >
> > > > > > This challenge in RL closely mirrors the issue in SFDA tasks, where label noise in pseudo-labels induces confirmation bias [e], [f] and degrades model performance during training time. Accordingly, from the perspective of deep-learning training, we have applied the exploration–exploitation strategy to SFDA by leveraging Tsallis mutual information (TMI). Our method dynamically fine-tunes the balance between exploration and exploitation via Tsallis entropy (TE) that adapts to the model’s learning state. This approach has not been addressed in prior work, and we believe it offers meaningful new insights for SFDA tasks.
> > > > > >
> > > > > > ---
> > > > > >
> > > > > > **References**
> > > > > >
> > > > > > [e] Pseudo-Labeling and Confirmation Bias in Deep Semi-Supervised Learning. Eric Arazo et al. IJCNN, 2020.
> > > > > >
> > > > > > [f] Two Wrongs Don’t Make a Right: Combating Confirmation Bias in Learning with Label Noise. Mingcai Chen et al. AAAI, 2023.

---

> > > > > > > ### Comment · Reviewer_9Bc5 · 2025-08-09
> > > > > > >
> > > > > > > The responses have addressed most of my concerns. I have no further questions, and I am raising the score to 5.

---

> > > > > > > > ### Author Response · Authors · 2025-08-09
> > > > > > > >
> > > > > > > > We sincerely appreciate your thoughtful follow-up and the favorable adjustment to the score. We will incorporate the relevant clarifications from our rebuttal into the final manuscript and make further revisions to enhance its clarity and completeness. Your careful review and guidance have been invaluable in strengthening this work. Thank you for your time and consideration.

---

### Official Review · Reviewer_PbfW · 2025-06-30

**Clarity:** 2
**Significance:** 3
**Originality:** 3
**Rating:** 4
**Confidence:** 3

**Summary:**

This paper introduces a two-stage, uncertainty-aware source-free domain adaptation (SFDA) method by leveraging an external, pre-trained vision-language model, CLIP. The authors alternate between fine-tuning the CLIP vision encoder and adapting the source model, incorporating Tsallis mutual information (TMI) to guide optimization. The proposed framework demonstrates improved SFDA performance and enhanced model calibration.

**Questions:**

1. Since the TMI loss is applied exclusively to the CLIP vision encoder, I am curious whether and how the improvements in CLIP representations contribute to the final adaptation performance of the target model. Could the authors provide further analysis on this aspect? Besides, is there any performance comparison between the fine-tuned CLIP encoder and the final adapted target model on the target domain?

2. What is the quality of the filtered hard pseudo labels? Would incorporating the mixed soft prediction $\hat{p}$ into the PLMatch framework further improve adaptation performance?

3. The authors mentioned the computational limitations of the proposed method. Are there any runtime or resource usage statistics to support its practical applicability?

4. Some notations and terminology can be clarified.

    - For instance, in Line 210, what does "EMA" stand for?

    - Also, the formulation of the TMI loss in Eq. (8) appears incorrect. The "min" operator may be confusing or misplaced, as the negative TMI is already used as the optimization objective.

**Ethical Concerns:**

["NO or VERY MINOR ethics concerns only"]

**Final Justification:**

During the rebuttal, the authors addressed most of my concerns and explicitly discussed the role of fine-tuning the external pre-trained vision-language model in SFDA. I believe these technical discussions and evaluations could benefit the community, so I decided to update my score.

**Limitations:**

The authors have mentioned their limitations in the Conclusion and limitations Section.

**Quality:**

3

**Strengths And Weaknesses:**

### **Strengths**

- **Novelty**
  The use of Tsallis mutual information for uncertainty-aware training captures model dynamics during adaptation. This approach appears novel and may offer valuable insights to the community.

- **Empirical Effectiveness**
  The proposed method improves performance on SFDA benchmarks. The experimental results further demonstrate that the adapted model is better calibrated under the uncertainty-aware training paradigm.

### **Weaknesses**

- **Insufficient Experimental Analysis**
  - The computational efficiency and scalability of the method are not sufficiently discussed. This raises concerns about the practical value of the proposed approach. Please refer to **Question 3** for more details.
  - There is no direct comparison with some recent state-of-the-art methods that also leverage pre-trained foundation models, such as [1].
  - I also have concerns regarding the overall completeness of the experimental section; please see **Questions 1 and 2** for specific points.

- **Methodological Design and Motivation**
  - The proposed method incorporates a pre-trained vision-language model (CLIP) as a complementary module within the SFDA framework while continuing to train the vision encoder of CLIP. This design choice introduces several questions: Why not use CLIP directly as the final target model? Could the CLIP vision encoder be better integrated into the target model? These concerns are further elaborated in **Question 1**.

- **Presentation and Clarity**
  - The overall presentation and mathematical formulations could be improved for better clarity and readability. Please refer to **Question 4** for specific suggestions.


**Reference**
[1] Proxy Denoising for Source-Free Domain Adaptation. Song Tang, et al. *ICLR 2025*.

---

> ### Author Rebuttal · Authors · 2025-07-31
>
> **Discussion on the use of CLIP**
> 1. *How the improvements in CLIP representations contribute to the final adaptation performance*
>
> In the proposed framework, fine-tuned CLIP contributes to performance improvement in two main aspects: **pseudo label refinement and knowledge distillation**.
>
> We begin by analyzing ① **how the frozen CLIP model contributes to target model adaptation performance**, and then ② **examine the additional benefits brought by fine-tuning CLIP representations** within our framework.
>
> For the ① analysis, we first compare two settings: (1) training the target model using our proposed framework without the frozen CLIP, and (2) training the target model with the frozen CLIP. This comparison quantifies how much the frozen CLIP improves the performance of the target model in our method.
>
> **Table G.** Performance comparison between target models trained with and without frozen CLIP on the Office-Home dataset.
>
> |Settings|→A|→C|→P|→R|Avg.|
> |-|-|-|-|-|-|
> |**Target w/o CLIP**|71.5|57.9|83.0|83.2|73.9|
> |**Target w/ CLIP**|**80.7**|**66.2**|**88.1**|**89.1**|**81.0**|
>
> However, as noted in Lines 87–89 of the main paper, the frozen CLIP model yields suboptimal representations for the target domain, underscoring the need for domain-specific adaptation. To address this (related to analysis ②), we fine-tune CLIP using TMI-based training.
>
> The improved CLIP features significantly enhance target model performance through **both pseudo label refinement and knowledge distillation**, as shown in **Table H**.
>
> **Table H.** Performance comparison between target models trained using the frozen CLIP and fine-tuned CLIP (ours) on the Office-Home dataset.
>
> |Settings|→A|→C|→P|→R|Avg.|
> |-|-|-|-|-|-|
> |**Target w/o CLIP-tuning**|80.7|66.2|88.1|89.1|81.0|
> |**Target w/ CLIP-tuning**|**83.3**|**73.8**|**90.8**|**91.0**|**84.7**|
> ---
> 2. *Why not use CLIP directly as the final target model? / Is there any performance comparison between the fine-tuned CLIP encoder and the final adapted target model on the target domain?*
>
> The primary objective of the proposed framework is to train the target model, not to directly use CLIP as the final model. The purpose of CLIP fine-tuning is therefore not to replace the target model, but rather to help the target model training with pseudo label refinement and knowledge distillation.
>
> As shown in **Table I**, the final target model (Proposed) outperforms both the frozen (CLIP w/o tuning) and fine-tuned CLIP models (CLIP w/ tuning), confirming that CLIP is best used as an auxiliary module rather than the main predictor.
>
> **Table I.** Performance comparison on the Office-Home dataset among the frozen CLIP, fine-tuned CLIP via the proposed method, and the final target model trained through our proposed framework.
>
> |Settings|A→C|A→P|A→R|C→A|C→P|C→R|P→A|P→C|P→R|R→A|R→C|R→P|Avg.|
> |-|-|-|-|-|-|-|-|-|-|-|-|-|-|
> |**CLIP w/o tuning**|60.1|84.2|85.8|75.2|84.2|85.8|75.2|60.1|85.8|75.2|60.1|84.2| 76.3|
> |**CLIP w/ tuning**|72.2|89.8|90.1|82.2|89.9|90.1|81.6|72.4|90.3|82.2|72.4|90.0|83.6|
> |**Proposed**|**73.6**|**90.4**|**91.0**|**83.6**|**90.7**|**90.9**|**82.7**|**73.7**|**91.2**|**83.6**|**74.0**|**91.2**|**84.7**|
> ---
> 3. *Could the CLIP vision encoder be better integrated into the target model?*
>
> In our framework, the CLIP encoder aligns visual features with text embeddings, while the target model aligns with classifier weights—reflecting fundamentally different objectives. This makes direct integration challenging, though exploring unified alignment strategies is a promising future direction.
>
> ---
> **Further analysis on Pseudo label**
> 1. *What is the quality of the filtered hard pseudo labels?*
>
> We conducted a quantitative analysis of pseudo label (PL) accuracy and sampling ratio per cycle on the Office-Home dataset [32]. % values indicate PL accuracy, and values in parentheses show the sampling ratio.
>
> While full results were obtained, we report four representative tasks due to space limits. PL accuracy declined across cycles (92.7%, 89.8%, 88.4%, 87.5%) as more PLs introduced inevitable noise. However, prior work [a], [b] shows that such noise can improve generalization. Similarly, our model (**Table E**, discussed in our response to **NoPH’s Question 2**) benefits from using more samples—even noisy ones—rather than strictly controlling PL accuracy.
>
> **Table J.** Cycle-wise PL accracy and sampling ratio across different tasks using the proposed method.
>
> |Task|Cycle 1|Cycle 2|Cycle 3|Cycle 4|
> |-|-|-|-|-|
> |→A|93.91%(0.51)|90.68%(0.78)|88.82%(0.87)|87.49%(0.91)|
> |→C|86.50%(0.40)|81.33%(0.72)|79.13%(0.84)|77.87%(0.90)|
> |→P|94.46%(0.67)|93.00%(0.90)|92.51%(0.89)|92.13%(0.90)|
> |→R|95.77%(0.67)|94.08%(0.88)|93.30%(0.94)|92.66%(0.96)|
>
> ---
> 2. *Would incorporating the mixed soft prediction $\mathcal{\hat{p}}$ into the PLMatch framework further improve adaptation performance?*
>
> We conducted additional experiments on the PLMatch framework using the mixed soft prediction $\mathcal{\hat{p}}$, applying both KL divergence and cross-entropy loss sequentially.
>
> **Table K.** Comparison of target model performance according to different label types and loss functions used in the PLMatch framework on the Office-Home dataset.
>
> |Label|Loss|→A|→C|→P|→R|Avg.|
> |-|-|-|-|-|-|-|
> |Soft |KL|82.7|72.5|90.6|90.6|84.1|
> |Soft |CE|83.0|73.4|90.6|90.8|84.4|
> |Hard|CE|**83.3**|**73.8**|**90.8**|**91.0**|**84.7**|
>
> While prior work [c] suggests that soft labels can be beneficial under label noise, our results show the opposite. We find that with high noise, soft labels may overly smooth targets and hinder convergence. Based on this interpretation and prior findings [d], we adopt a cross-entropy loss with hard pseudo labels in the PLMatch framework.
>
> ---
> **Computational limitations**
>
> To provide a clearer analysis of computational limitations, we report training time per batch (TTB), inference time (IT) per sample and GPU memory usage for the Art→Clipart task from the Office-Home dataset [32].
>
> **Table L.** Comparison of Computational Cost among SFDA Methods
>
> | Method | TTB ↓ (s) | IT ↓ (ms) | GPU memory ↓ (GB) |
> |-|-|-|-|
> |SHOT|0.3621|0.608|7.759|
> |DIFO|0.6576|0.608|8.340|
> |Ours| 0.5238 |0.608|14.452|
>
> Although our method requires higher GPU memory due to using both CLIP and the target model, its hierarchical structure—with separate cycle- and iteration-level processes—enables faster training than DIFO, which lacks such decomposition. Inference time per sample is identical across methods, as it depends only on the target model.
>
> ---
> **SOTA comparison**
>
> Thank you for your valuable observation. We have carefully examined the recent method (ProDe) you have referenced. Despite the emergence of new methods, our method still achieves state-of-the-art (SOTA) performance on two out of the three benchmark datasets (Office-Home [32] and VisDA-C [20]).
>
> **Table M.** Comparison of target model performance between recent method (ProDe) and ours on Office-Home dataset. The asterisk (*) indicates the performance reported in the referenced paper.
>
> |Method|A→C|A→P|A→R|C→A|C→P|C→R|P→A|P→C|P→R|R→A|R→C|R→P|Avg.|
> |-|-|-|-|-|-|-|-|-|-|-|-|-|-|
> |ProDe*|72.7|**92.3**|90.5|82.5|**91.5**|90.7|82.5|72.5|90.8|83.0|72.6|**92.2**|84.5|
> |Ours|**73.6**|90.4|**91.0**|**83.4**|90.7|**90.9**|**82.7**|**73.7**|**91.2**|**83.6**|**74.0**|91.2|**84.7**|
>
> **Table N.** Comparison of target model performance between recent method (ProDe) and ours on VisDA-C dataset.
> |Method|plane|bike|bus|car|horse|knife|mcycle|person|plant|sktbrd|train|truck|Avg.|
> |-|-|-|-|-|-|-|-|-|-|-|-|-|-|
> |ProDe*|98.3|**92.4**|86.6|**80.5**|98.1|**98.0**|92.3|84.3|94.7|97.0|94.1|**75.6**|91.0|
> |Ours|**98.6**|91.2|**89.8**|79.8|**98.1**|97.8|**95.0**|**84.7**|**96.2**|**97.2**|**95.0**|73.1|**91.4**|
>
> For the remaining dataset, DomainNet-126 [19], we faithfully followed the official implementation provided by the referenced paper, including the pre-trained source model weights, codebase, and experimental settings. To account for potential performance variability, we conducted seven rounds of experiments, carefully varying the number of training epochs, random seeds, and other configurations. The asterisk (*) indicates the performance reported in the referenced paper.
>
> **Table O.** Comparison of target model performance between recent method (ProDe) and ours on the DomainNet-126 dataset.
> | Method|C→P|C→R|C→S|P→C|P→R|P→S|R→C|R→P|R→S|S→C|S→P |S→R |Avg.|
> |-|-|-|-|-|-|-|-|-|-|-|-|-|-|
> | ProDe(10epochs)|76.47 ± 0.61|87.50 ± 0.53|73.17 ± 1.04|81.73 ± 0.70|87.97 ± 0.15|73.00 ± 0.10|81.10 ± 0.30|76.60 ± 0.26|73.97 ± 0.45|79.13 ± 0.55|76.13 ± 0.84|88.10 ± 0.36|79.57 ± 0.25|
> | ProDe(15epochs)|76.1|88.4|72.4|81.1|88.1|74.0|80.5|75.2|73.1|79.6|75.6|87.4|79.3|
> | ProDe(30epochs)|75.1|87.7|72.9|80.7|87.2|71.7|79.8|74.3|73.2|80.0|75.4|87.7|78.8|
> | ProDe*|83.2|92.4|79.0|85.0|92.3|79.3|85.5|83.1|79.1|85.5|83.4|91.0|84.9|
> | Ours|79.9|89.5|79.0|82.8|89.7|79.3|83.0|80.7|78.9|83.5|80.3|89.5|83.0|
>
> Specifically, while the referenced method uses 15 epochs as its default training setting, firstly we also tested at its default training setting, without any changes. Subsequently, we tested alternative setups with fewer epochs (10) and more epochs (30). After identifying that the lower-epoch setting generally yielded better average performance, We further repeated the experiments with different random seeds. Despite these extensive efforts, we were ultimately unable to reproduce the performance reported in the original paper.
>
> ---
>
> **References**
>
> [a] FlexMatch: Boosting Semi-Supervised Learning with Curriculum Pseudo Labeling. Bowen Zhang et al. NeurIPS 2021.
>
> [b] Boosting Semi-Supervised Learning by Bridging High and Low-Confidence Predictions. Khanh-Binh Nguyen et al. ICCV 2023.
>
> [c] Does Label Smoothing Mitigate Label Noise?. Michal Lukasik et al. ICML 2020.
>
> [d] To Smooth or Not? When Label Smoothing Meets Noisy Labels. Jiaheng Wei et al. ICML 2022.

---

> > ### Comment · Reviewer_PbfW · 2025-08-04
> >
> > I have carefully read the authors’ rebuttal, the other reviewers' comments, and the corresponding responses. I sincerely appreciate the authors' efforts during the rebuttal phase, especially the additional experimental results addressing many of my concerns, such as the role of CLIP and its update throughout the adaptation process, as well as the evolution of pseudo-label quality.
> >
> > I strongly encourage the authors to include in the main paper the insightful discussions provided in the rebuttal (such as the pseudo-label quality and the comparison between different cycle-wise and iteration-wise update rounds, as mentioned in response to Reviewer EPpg), as they are both technically meaningful and practically relevant.
> >
> > I acknowledge the technical contribution of adapting the CLIP vision encoder to concrete downstream tasks and leveraging its updated output to improve the adaptation process. However, I remain concerned about (1) the relatively marginal contribution of the TMI loss, as shown in Table 3 and in response to Reviewer NoPH (which is claimed as one of the main contributions), and (2) the clarity and presentation issues in the current manuscript. Therefore, I tend to maintain my original score.

---

> > > ### Author Response · Authors · 2025-08-05
> > >
> > > Once again, thank you for your review and in-depth feedback. We will ensure that all key experiments discussed in the rebuttal are incorporated into the final manuscript with enhanced clarity and presentation. Although TMI is a vital component, our core strategy lies in co-training the target model with CLIP. We would like to highlight our contributions as follows:
> > >
> > > (1) **Dual-facet pseudo labeling**, which refines labels by enforcing agreement between CLIP and the task-specific target model without complex pseudo labeling algorithms. (2) **Asymmetric iteration design** (as discussed in the response of **W3** from Reviewer **NoPH**), which accelerates model convergence by focusing the current reliable pseudo label on each cycle. (3) **Uncertainty-based Tsallis mutual information**, which guides the SFDA process to balance exploration during initial training phases and exploitation in later stages.
> > >
> > > The proposed approach provides superior results over existing techniques (The table below lists the two best existing techniques as benchmark methods) based on the above novelties as following table.
> > >
> > > | Method | Office-Home [32] | VisDA-C [20] | DomainNet-126 [19] |
> > > |-|-|-|-|
> > > | GKD [26] | 72.4 | 82.6 | 68.7|
> > > | DIFO [27] | 83.2 | 89.9 | 80.0 |
> > > | Ours | **84.7 (+1.5%)** | **91.4 (+1.5%)** | **83.0 (+3.0%)** |
> > >
> > > As shown in the above table, the proposed method achieves **+1.5%~3.0% improvements over the existing SOTA methods on average for three datasets.**
> > >
> > > To check the robustness of TMI, we extended our experiments to the PACS datasets [a], known for its pronounced domain shifts. As shown in table below, with ResNet-50  as the CLIP vision encoder, TMI still yields a **+0.3%** performance enhancement over MI—a gain that is especially meaningful given ViT’s already strong baseline performance.
> > >
> > > |Component|A→C|A→P|A→S|C→A|C→P|C→S|P→A|P→C|P→S|S→A|S→C|S→P|Avg.|
> > > |-|-|-|-|-|-|-|-|-|-|-|-|-|-|
> > > |Source|58.8|97.9|44.4|61.9|81.4|54.7|65.6|26.3|28.6|28.9|54.4|33.7|53.1|
> > > |Ours (w/ MI)|96.8|99.5|**89.6**|**95.8**|99.3|**89.8**|95.5|**96.5**|86.7|95.0|94.8|99.3|94.9|
> > > |Ours (w/ TMI)|**96.9**|**99.5**|88.9|95.6|**99.6**|89.6|**95.5**|96.2|**89.0**|**95.5**|**96.4**|**99.5**|**95.2**|
> > >
> > > Across all our rebuttal experiments, **we observed an average improvement of +0.2% with ViT/B-32 and +0.4% with ResNet backbones**. (In SFDA, overall performance gains under 1.0% are generally regarded as meaningful enhancements [1], [26], [36], underlining the significance of this result.) We believe that these differences stem from ViT’s stronger inductive biases and robustness compared to convolutional archiectures [b], [c]. Given CLIP’s versatility across backbones, we argue that TMI-based fine-tuning is generally the preferable approach for SFDA.
> > >
> > > Thank you once again for your time and valuable feedback. We will incorporate your feedback to significantly enhance both our contributions and the way we present them.
> > >
> > > ---
> > > **References**
> > >
> > > [a] Deeper, Broader and Artier Domain Generalization. Da Li et al. ICCV, 2017.
> > >
> > > [b] Vision Transformers Are Robust Learners. Sayak Paul and Pin-Yu Chen. AAAI, 2022.
> > >
> > > [c] ConvNet vs Transformer, Supervised vs CLIP: Beyond ImageNet Accuracy. Kirill Vishniakov et al. arXiv:2311.09215, 2023.

---

> > > > ### Comment · Reviewer_PbfW · 2025-08-06
> > > >
> > > > Thanks to the authors for the further explanation. I have updated my score and hope the authors will revise the final manuscript to clearly clarify their contributions.

---

> > > > > ### Author Response · Authors · 2025-08-06
> > > > >
> > > > > We truly appreciate your encouragement to strengthen the clarity of our manuscript. We will be sure to integrate the insightful discussions from our rebuttal directly into the final manuscript. Your guidance has been invaluable, and we are committed to revising the final manuscript to clearly highlight these technically meaningful and practically relevant contributions.

---

### Official Review · Reviewer_NoPH · 2025-07-03

**Clarity:** 2
**Significance:** 3
**Originality:** 2
**Rating:** 4
**Confidence:** 3

**Summary:**

This paper addresses the problem that prior methods in Source-Free Domain Adaptation (SFDA) rely on inaccurate pseudo-labels in the early stages of training, and the limited task-specific adaptability of CLIP-based prompt tuning due to its frozen vision encoder, which results in low-quality image representations. To solve this problem, the authors propose assigning pseudo labels only to samples where the target model and the CLIP model agree, while utilizing consistency learning for the remaining samples to enhance feature representation. Additionally, they propose an uncertainty-aware optimization strategy based on Tsallis mutual information (TMI) that dynamically aligns the CLIP vision encoder to the target by balancing exploration and exploitation. The proposed method outperforms existing baselines on standard SFDA benchmarks.

**Questions:**

1. It would be helpful if the Related Work section included a dedicated paragraph on SFDA
2. In Figure 1, why does the pseudo-label accuracy of the proposed method drop in later iterations? Is it because the number of assigned pseudo labels increases? If so, could the trade-off between accuracy and the number of pseudo labels assigned be controlled? And under what conditions would this be most efficient?
3. In Figure 3, specifically in the CPG part, what is the reason for the varying bar widths?
4. Why was the averaging factor set to 2 in the prediction mixture? Have you experimented with other weighting schemes?
5. The paper states that consistency training is applied only to the remaining samples, which are not assigned pseudo labels. However, Equation (10) computes the loss over all N samples. Shouldn't it be restricted to the unlabeled subset? This discrepancy needs clarification.

**Ethical Concerns:**

["NO or VERY MINOR ethics concerns only"]

**Final Justification:**

The concerns I raised were adequately addressed by further clarification and experimentation through the authors’ rebuttal. I particularly appreciate the responses to my two main concerns:

1) While the performance gains with TMI are marginal, I acknowledge that it improves various aspects, such as calibration and label quality.

2) The explanation of the asymmetric use of uncertainty modeling in the alignment mechanism clarifies the design intent and rationale for the framework.

Finally, this paper presents a well-structured motivation and a corresponding method. However, the performance gain of the core method, TMI, is very marginal, and issues such as the figure mentioned in Q3 and the inconsistency between the equation and description noted in Q5 raise concerns about the overall quality of the paper. Therefore, I have decided to assign a borderline accept.

**Limitations:**

The limitations of the proposed method are described in Section 5 (Conclusion and Limitations).

**Paper Formatting Concerns:**

No major formatting issues were found. The paper appears to follow the NeurIPS 2025 formatting guidelines.

**Quality:**

2

**Strengths And Weaknesses:**

Strengths
1. This paper clearly highlights the limitations of prior methods and motivates the proposed approach through insightful experiments, particularly by visualizing the weaknesses of DIFO and the frozen CLIP encoder in Figures 1 and 2.

Weaknesses
1. The descriptions in the Introduction and Method sections are not well aligned, which hinders comprehension. While the Introduction briefly outlines only Dual-facet pseudo labeling and TMI-based vision optimization, the Method section introduces several new components—such as the Cycle-level process / Iteration-level process framework, CPG, DVO, and PLMatch—without prior context. As a result, it is unclear whether these components represent core contributions or simply implementation details.

2. As shown in Table 3, the performance gain of TMI, which is one of the main contributions of the paper, is marginal compared to the standard MI. Considering that TMI introduces additional computational cost (e.g., the calculation of ψ), the benefit of using TMI may not be significant in practice.

3. Both TMI in the Cycle-level process and CLIP-guided knowledge distillation in the Iteration-level process appear to align the CLIP model with the target model. However, only TMI incorporates adaptation uncertainty (ψ), while CL does not. The paper lacks a clear explanation as to why uncertainty modeling is applied to one but not the other, and further clarification on the design intention behind this asymmetry would be helpful.

---

> ### Author Rebuttal · Authors · 2025-07-31
>
> **(W1) Implementation details and core contributions**
>
> We sincerely apologize for the confusion caused by the misalignment between the Introduction and Method sections.
> We will clearly distinguish between the implementation details and core contributions as follows:
>
> | Implementation details | Core contributions |
> |-|-|
> |Cycle-level process|CPG, DVO|
> |Iteration-level process|PLMatch|
>
> Specifically, the concepts of the cycle-level and iteration-level processes are introduced to clearly articulate the hierarchical structure of our core contributions. In the cycle-level process, computationally intensive components such as the proposed CPG and DVO are executed. In contrast, the iteration-level process applies to the relatively lightweight proposed PLMatch module.
>
> ---
> **(W2) Marginal performance gain of TMI compared to MI**
>
> We acknowledge that the comparison of the performance gain between TMI and MI reported in Table 3 may appear marginal. To address this concern, we conducted an ablation study comparing MI and TMI using three different seeds. (For transparency, we note that the TMI results are identical to those used in the statistical rigor analysis in **Reviewer EPpg's W1**.)
> Additionally, following prior work [27], we compared the performances of the proposed method with MI and TMI when the CLIP vision encoder is replaced with ResNet-50 and ResNet-101 as backbones.
>
> As shown in **Table A**, the performance advantage of TMI over MI consistently appears, confirming the findings in Table 3 of the main text. Notably, the performance gap is more pronounced when using lightweight backbones such as ResNet variants, compared to the ViT-B/32 used in the final model. This suggests that TMI provides greater performance gains over MI in scenarios involving lightweight architectures.
>
> **Table A.** An ablation study on MI and TMI using the Office-Home dataset.
> |Backbone|Method|A→C|A→P|A→R|C→A|C→P|C→R|P→A|P→C|P→R|R→A|R→C|R→P|Avg.|
> |-|-|-|-|-|-|-|-|-|-|-|-|-|-|-|
> |ViT-B/32|Ours (w/MI)|73.07 ± 0.15|**90.47 ± 0.21**|90.80 ± 0.1|83.27 ± 0.06|90.50 ± 0.17|**90.93 ± 0.12**|82.70 ± 0.0|73.03 ± 0.12|**91.00 ± 0.17**|83.70 ± 0.1|73.17 ± 0.32|91.13 ± 0.06|84.47 ± 0.12|
> ||Ours (w/TMI)|**73.27 ± 0.31**|90.43 ± 0.06|**90.87 ± 0.12**|**83.43 ± 0.15**|**90.73 ± 0.06**|90.83 ± 0.06|**83.00 ± 0.26**|**73.33 ± 0.35**|90.93 ± 0.25|**83.73 ± 0.15**|**73.77 ± 0.25**|**91.20 ± 0.10**|**84.63 ± 0.08**|
> |RN50|Ours (w/MI)|59.0|84.8|86.4|**78.7**|85.5|**86.8**|78.7|60.3|**87.2**|**78.7**|60.4|**85.2**|77.6|
> | |Ours (w/TMI)|**60.5**|**84.9**|**86.9**|78.5|**85.7**|86.7|**79.0**|**61.6**|86.9|**78.7**|**61.5**|**85.2**|**78.0**|
> |RN101|Ours (w/MI)|63.9|89.1|88.4|82.1|88.9|88.5|**82.5**|64.6|**89.1**|**82.7**|63.6|89.0|81.0|
> | |Ours (w/TMI)|**64.2**|**89.3**|**88.8**|**82.8**|**89.7**|**88.8**|81.7|**65.2**|89.0|**82.7**|**65.3**|**89.7**|**81.4**|
>
> Moreover, CLIP in our framework contributes in two major ways: knowledge distillation and pseudo label refinement. To assess TMI's contribution in these areas, we measured Expected Calibration Error (ECE) of CLIP, CLIP accuracy (CLIP Acc.), and pseudo label accuracy (PL Acc.). As shown in **Table B**, the results suggest that CLIP trained with TMI achieves a more stable and well-calibrated learning state than with MI, which positively affects the quality of the pseudo labels. These findings indicate that TMI-based training can be especially useful in scenarios with large domain gaps.
>
> **Table B.** Analysis of fine-tuned CLIP with MI and TMI.
>
> |Task|Loss in DVO|ECE(↓)|CLIP Acc.(↑)|PL Acc.(↑)|
> |-|-|-|-|-|
> |A→C|MI|0.20|77.0|71.8|
> ||TMI|**0.18**|**77.4**|**72.2**|
> |P→C|MI|0.20|76.9|71.6|
> ||TMI|**0.18**|**77.7**|**72.4**|
> |R→C|MI|0.19|77.6|71.6|
> ||TMI|**0.17**|**77.7**|**72.4**|
> ---
> **(W3) Asymmetric use of uncertainty modeling in alignment mechanisms**
>
> To clarify our design intentions, we provide the following explanation:
>
> At the cycle level, DVO with TMI encourages informed updates through uncertainty-driven exploration, resulting in slower but more deliberate convergence. In contrast, the iteration-level process focuses on rapid convergence by leveraging confident pseudo-labeled samples, avoiding uncertainty-based strategies and instead adopting knowledge distillation.
>
> To assess the impact of this design, we extended TMI-based objectives to the iteration level. As shown in **Table C**, our full design (Asymmetry) consistently outperforms the TMI-only variant (Symmetry), demonstrating its effectiveness.
>
> **Table C.** Comparison between symmetric and the proposed asymmetric approach on the Office-Home dataset.
>
> | Settings |Objectives in cycle-level|Objectives in iteration-level|→A|→C|→P|→R|Avg.|
> |-|-|-|-|-|-|-|-|
> |Symmetry|TMI|TMI|78.1|69.1|86.9|86.9|80.3|
> |Asymmetry (Proposed)|TMI|KL|**83.3**|**73.8**|**90.8**|**91.0**|**84.7**|
>
> ---
> **(Q1) Lack of dedicated discussion on SFDA in Related Work**
>
> Thank you for your valuable suggestion. We will add a new subsection to briefly introduce representative prior works that inspired our method—such as [11], [1], [27]—to help readers better understand the motivation and positioning of our approach.
>
> ---
> **(Q2) Trade-off between accuracy and the number of pseudo labels**
>
> We appreciate your observation. As correctly pointed out in your question, the drop in pseudo-label accuracy during later iterations is indeed attributed to the increasing number of assigned pseudo labels (PLs), which highlights a natural trade-off between label quality and coverage. To further analyze this trade-off, we conducted additional experiments on the Office-Home dataset [32] using confidence thresholding (CT) to assign PLs, controlling the balance between PL accuracy and quantity.
>
> **Table D.** Comparison of PL accuracy and sampling ratio across cycles for different CT strategies and our proposed method on the Office-Home dataset.
> |Threshold|Scenario|Cycle 1|Cycle 2|Cycle 3|Cycle 4|
> |-|-|-|-|-|-|
> |0.85|P→C|98.82%(0.03)|97.19%(0.20)|94.72%(0.37)|93.07%(0.47)|
> ||R→A|99.18%(0.15)|98.67%(0.34)|98.09%(0.45)|97.88%(0.52)|
> |0.95|P→C|100%(0.007)|98.79%(0.09)|97.41%(0.19)|96.70%(0.28)|
> ||R→A|99.03%(0.04)|99.54%(0.18)|99.39%(0.27)|99.49%(0.32)|
> |0.85 to 0.95|P→C|98.82%(0.04)|98.32%(0.18)|96.49%(0.29)|96.11%(0.34)|
> ||R→A|99.18%(0.15)|98.64%(0.30)|98.67%(0.37)|98.75%(0.40)|
> |Ours|P→C|87.85%(0.37)|81.56%(0.71)|79.44%(0.84)|78.22%(0.90)|
> ||R→A|93.58%(0.60)|90.71%(0.79)|88.91%(0.87)|87.62%(0.91)|
>
> **Table E.** Comparison of classification performance between different CT strategies and the proposed method on the Office-Home dataset.
> |Threshold|A→C|A→P|A→R|C→A|C→P|C→R|P→A|P→C|P→R|R→A|R→C|R→P|Avg.|
> |-|-|-|-|-|-|-|-|-|-|-|-|-|-|
> |0.85|73.1|90.3|90.6|82.9|90.5|90.8|82.4|72.8|91.0|**83.8**|73.8|90.9|84.4|
> |0.95|72.9|**90.5**|90.8|82.7|90.4|90.6|82.3|73.0|90.8|83.7|73.5|91.0|84.3|
> |0.85 to 0.95|72.9|90.4|90.7|82.8|90.5|90.6|82.3|72.7|90.9|83.6|73.6|91.0|84.3|
> |Ours|**73.6**|90.4|**91.0**|**83.6**|**90.7**|**90.9**|**82.7**|**73.7**|**91.2**| 83.6|**74.0**|**91.2**|**84.7**|
>
> The rows labeled “0.85” and “0.95” in **Table D** correspond to settings where a fixed CT of 0.85 or 0.95 was applied throughout training.
>
> The row “0.85 to 0.95” indicates a linearly increasing threshold, designed to progressively raise the confidence threshold in order to maintain PL accuracy across cycles.
>
> As shown in **Table D**, CT-based methods produce high-quality pseudo labels but label fewer samples than ours. In contrast, our approach achieves better overall performance and more stable training (**Table E**). This aligns with the training strategy [a] of gradually incorporating lower confidence labels. This enables broader learning and mitigates overfitting, suggesting that our method ensures practical and stable label quality without complex thresholds or strategies.
>
> ---
> **(Q3) Figure modification**
>
> We sincerely apologize for any confusion in Figure  3. Although it was initially intended to emphasize class separability along with color, we will unify the bar widths in the final version.
>
> ---
> **(Q4) Justification for averaging factor and weighting schemes in prediction mixture**
>
> Thank you for your insightful question. We empirically set the averaging factor used in the prediction mixture to 2 after experimenting with various schemes.
>
> **Table F.** Ablation study on different weighting schemes used for the prediction mixture on the Office-Home dataset.
> |Scheme|**α**|**(1 - α)**|**→A**|**→C**|**→P**|**→R**|**Avg.** |
> |-|-|-|-|-|-|-|-|
> |Fixed|0.1|0.9|82.5|73.6|90.6|90.7|84.6|
> |-|0.3|0.7|83.0|73.4|90.7|90.8|84.5|
> |-|0.5|0.5|83.3|**73.8**|**90.8**|**91.0**|**84.7**|
> |-|0.7|0.3|**83.4**|73.5|90.7|90.8|84.6|
> |-|0.9|0.1|83.3|72.7|90.6|90.7|84.3|
> |Dynamic|Linear||83.0|73.2|90.4|90.8|84.3|
> |-|Random||83.1|73.0|90.6|90.9|84.4|
> |-|Exponential||82.9|73.1|90.4|90.8|84.3|
>
> We conducted experiments using both fixed and dynamic balancing schemes for the prediction mixture. In this context, we denote α as the weight assigned to the target model's prediction, and (1 − α) as the weight for CLIP's prediction. Additionally, we explored the following dynamic strategies:
>
> Linear and Exponential: α increases linearly or exponentially over cycles
>
> Random: α is sampled randomly from a uniform distribution
>
> Through these experiments, we found that the fixed averaging factor (α = 0.5) provided a simple yet effective solution and served as a strong baseline throughout our work.
>
> ---
> **(Q5) Inconsistency between textual description and Equation (10) regarding consistency training**
>
> We sincerely thank the reviewer for pointing out the discrepancy between Eq. (10) and the description in Lines 240–242. As correctly reflected in Eq. (10), we apply consistency training to all samples regardless of their pseudo-label assignment. We will revise the sentence in the final version to more accurately reflect our implementation and design intent.
>
> ---
> **References**
>
> [a] Tri-Net for Semi-Supervised Deep Learning. Dong-Dong Chen et al. IJCAI 2018.

---

> > ### Comment · Reviewer_NoPH · 2025-08-05
> >
> > I appreciate the author's thoughtful and detailed rebuttal.
> >
> > The concerns I raised were adequately addressed by further clarification and experimentation by the authors. I particularly appreciate the responses to my two main concerns:
> >
> > 1) While the performance gains with TMI are marginal, I acknowledge that it improves various aspects, such as calibration and label quality.
> >
> > 2) The explanation of the asymmetric use of uncertainty modeling in the alignment mechanism clarifies the design intent and rationale for the framework.
> >
> > However, I still think the performance improvements for some methods are marginal and not enough to warrant a high score.

---

> > > ### Author Response · Authors · 2025-08-05
> > >
> > > We sincerely appreciate your careful review of the counterargument and your positive assessment of the key concerns. In addition to TMI, the proposed approach provides superior results over existing techniques (The table below lists the two best existing techniques as benchmark methods) based on novelties described below. (1) **Dual-facet pseudo labeling**, which efficiently calibrates pseudo labels by requiring agreement between CLIP and the target model without complex algorithms. (2) **An asymmetric design** that accelerates model convergence by focusing on the reliable pseudo labels on each cycle. (3) **Uncertainty-based Tsallis mutual information**, offering insight into the necessity of exploration in the early stages and exploitation in the later stages of SFDA training.
> > >
> > > | Method | Office-Home [32] | VisDA-C [20] | DomainNet-126 [19] |
> > > |-|-|-|-|
> > > | GKD [26] | 72.4 | 82.6 | 68.7|
> > > | DIFO [27] | 83.2 | 89.9 | 80.0 |
> > > | Ours | **84.7 (+1.5%)** | **91.4 (+1.5%)** | **83.0 (+3.0%)** |
> > >
> > > As shown in the above table, the **proposed method achieves +1.5%~3.0% improvements over the existing SOTA methods** on average for three datasets.
> > > To confirm the robustness of TMI, we conducted further experiments on the PACS dataset [a], which exhibits larger domain gaps than Office-Home [32]. When using ResNet-50 as the CLIP vision encoder, TMI achieves a **+0.3%** improvement over MI. **In SFDA, overall performance gains under 1.0% are generally regarded as meaningful enhancements [1], [26], [36], underlining the significance of this result.**
> > >
> > > |Component|A→C|A→P|A→S|C→A|C→P|C→S|P→A|P→C|P→S|S→A|S→C|S→P|Avg.|
> > > |-|-|-|-|-|-|-|-|-|-|-|-|-|-|
> > > |Source|58.8|97.9|44.4|61.9|81.4|54.7|65.6|26.3|28.6|28.9|54.4|33.7|53.1|
> > > |Ours (w/ MI)|96.8|99.5|**89.6**|**95.8**|99.3|**89.8**|95.5|**96.5**|86.7|95.0|94.8|99.3|94.9|
> > > |Ours (w/ TMI)|**96.9**|**99.5**|88.9|95.6|**99.6**|89.6|**95.5**|96.2|**89.0**|**95.5**|**96.4**|**99.5**|**95.2**|
> > >
> > > Overall, across all rebuttal experiments, **we observe a +0.2% gain with ViT/B-32 backbones and a +0.4% gain with ResNet-based backbones**. We attribute these backbone-dependent differences to ViT’s stronger inductive biases and inherent robustness compared to convolutional architectures [b], [c]. Given CLIP’s flexibility to support multiple backbones, we believe that fine-tuning CLIP with TMI is, on average, the more advantageous choice for SFDA tasks. We respectfully ask you to consider these comprehensive results in your evaluation.
> > >
> > > Once again, we are immensely grateful for your constructive feedback, which has significantly strengthened our paper, and we thank you sincerely for taking time to read this detailed response.
> > >
> > > ---
> > > **References**
> > >
> > > [a] Deeper, Broader and Artier Domain Generalization. Da Li et al. ICCV, 2017.
> > >
> > > [b] Vision Transformers Are Robust Learners. Sayak Paul and Pin-Yu Chen. AAAI, 2022.
> > >
> > > [c] ConvNet vs Transformer, Supervised vs CLIP: Beyond ImageNet Accuracy. Kirill Vishniakov et al. arXiv:2311.09215, 2023.

---

> > > > ### Comment · Reviewer_NoPH · 2025-08-06
> > > >
> > > > Thank you for the additional experiments.  I apologize for not making it clear in my previous comment that I had already raised the score. In fact, I had slightly raised it at that time in recognition of the proposed method’s contributions, although I did not increase it significantly due to the small performance gap. Therefore, I will maintain the previously updated score.

---

> > > > > ### Author Response · Authors · 2025-08-06
> > > > >
> > > > > Thank you very much for the clarification and for the score update. We appreciate your recognition of our method's contributions and understand your decision to maintain the previously adjusted score. Your feedback is invaluable, and we will incorporate all of your comments into our final manuscript.

---

### Note · Authors · 2025-08-13

We sincerely thank the Reviewers and the Area Chair for their meticulous assessment and thoughtful understanding throughout the rebuttal period, which greatly helped refine our manuscript. During rebuttal, we addressed all concerns through a series of additional experiments and theory-grounded analyses, and incorporated all corresponding revisions into the final manuscript.

---

### **Dual-facet (perspective) pseudo labeling**

- Assign pseudo labels only when CLIP (domain-invariant perspective) and the target model (task-specific perspective) agree

**[What we proved]**
- Performance comparisons against existing pseudo labeling schemes (e.g., confidence thresholding)
- An analysis of the **trade-off** between pseudo label **quality** and **sampling ratio**

→ Even **without** complex algorithms, the proposed approach yields **high-quality pseudo labels** that lead to **performance gains**.

---

### **Exploration–exploitation strategy with the proposed Tsallis Mutual Information (TMI)**

- A novel loss that adapts to model uncertainty, **promoting exploration** early in training to mitigate confirmation bias, and **shifting to exploitation** later to leverage reliable predictions.

**[What we proved]**
- Robustness and consistent improvements **across various datasets and vision-encoder backbones**
- An analysis of TMI within our framework via **expected calibration error (ECE)**, **accuracy** of CLIP trained with TMI, and **pseudo label accuracy**

→ TMI delivers **significant gains** for both the target model and CLIP across datasets/backbones, and the staged exploration–exploitation scheme offers **novel** insights for SFDA.

---

### **Justification for the structural design**

**[What we proved]**
- The final **target model outperforms** the fine-tuned CLIP.
- Ablations with **frozen CLIP** and **fine-tuned CLIP** within our framework clarify **why CLIP is needed** and quantify its contribution.
- Hard vs. soft pseudo labels: results justify preferring hard pseudo labels for target model training.
- Comparisons of **soft pseudo label** generation mechanisms for CLIP vision optimization

---

### **Clarity of the manuscript**

- We clarified the intent behind our contributions and design choices.
- We added clearer explanations of the **hierarchical training scheme** and of **Tsallis Mutual Information (TMI)**.

---

Once again, we sincerely thank the Reviewers and the Area Chair for their valuable insights and comments.

---

### Decision · Program_Chairs · 2025-09-17

**Decision:**

Accept (poster)

**Comment:**

Dear authors,

Thank you for submitting draft. This draft has received overall positive reviews, including 1 Accept (Rating-5). We strongly suggest that authors should incorporate suggestions and elements presented during discussion and rebuttal in the final draft.

Congratulations.

regards
AC